



**Atmospheric Water-Soluble Organic Nitrogen (WSON) in the Eastern Mediterranean:**
**Origin and Ramifications Regarding Marine Productivity**
**Münevver Nehir[1] and Mustafa Koçak[1*]**
[1] Institute of Marine Sciences, Middle East Technical University, P.O. Box 28, 33731,
Erdemli-Mersin, Turkey
[*]Corresponding author (Tel: +90324-5213434; Fax: +90324-5212327;
mkocak@ims.metu.edu.tr
**Abstract**
Two-sized aerosol and rain sampling were carried out at a rural site located on the coast of the
Eastern Mediterranean, Erdemli, Turkey (36° 33′ 54″ N and 34° 15′ 18″ E). A total of 674
aerosol samples in two size fraction (coarse = 337; fine = 337) and 23 rain samples were
collected between March 2014 and April 2015. Samples were analyzed for $NO_3^-$, $NH_4^+$ and
ancillary water-soluble ions by Ion Chromatography and water-soluble total nitrogen (WSTN)
by applying a High Temperature Combustion Method. The mean aerosol WSON was 23.8 ±
16.3 nmol N m$^{-3}$, reaching a maximum of 79 nmol N m$^{-3}$, with about 66 % being associated
with coarse particles. The volume weighted mean (VWM) concentration of WSON in rain
was 21.5 µmol N L$^{-1}$. The WSON contributed 37 % and 29 % to the WSTN in aerosol and
rainwater, respectively.  Aerosol WSON concentrations exhibited large temporal variations
mainly due to rain and the origin of air mass flow. The highest mean aerosol WSON
concentration was observed in the summer and was attributed to the absence of rain and re-
suspension of cultivated soil in the region. The mean concentration of WSON during dust
events (38.2±17.5 nmol N m$^{-3}$) was 1.3 times higher than that of non-dust events (29.4±13.9
nmol N m$^{-3}$). Source apportionment analysis demonstrated that WSON was originated from
agricultural activities (43 %), secondary aerosol (20 %), nitrate (22 %), crustal (10 %) and
sea-salt (5 %). The dry and wet depositions of WSON were equivalent and amounted to 36 %
of the total atmospheric WTSN flux. Considering the Cilician Basin, the atmospheric water-
soluble nitrogen flux would sustain 33 % and 76 % of the new production in the associated
coastal and open waters, respectively.
**Keywords:** Atmospheric water-soluble organic nitrogen, mineral dust, source apportionment,
atmospheric deposition and marine productivity, Eastern Mediterranean



## 1. Introduction

Research assessing the atmospheric deposition of nitrogen (with a focus on inorganic
N in rainwater i.e. ammonium and nitrate) can be traced back to the mid-1800s (Miller, 1995
and references therein) as it was accepted to be a vital plant nutrient. Miller (1905) mentioned
about organic nitrogen in rain samples as well. To quote Miller: '*With regard to the amount of*
*organic nitrogen in the rainwater, the only available analyses relating to Rothamsted are*
*those of Frankland who found from 0.03 to 0.66 per million in 69 samples*'. Cornell et al.,
(1995) highlighted the importance of organic nitrogen in rain and snow accounting for almost
half of the total atmospheric dissolved nitrogen deposition. Since then, research defining the
quantitative importance of soluble organic nitrogen in the atmospheric transport of nitrogen
has greatly expanded (Neff et al, 2002; Cornell et al., 2003; Mace et al., 2003a, b, c; Gilbert et
al., 2005; Sorooshian et al., 2008; Violaki and Mihalopoulos, 2010; Violaki et al., 2010;
Altieri et al., 2016).
WSON arises from a variety of sources including both natural and anthropogenic.
Anthropogenic sources include agricultural activities (including fertilizer application, animal
husbandry), high temperature fossil fuel combustion, man-made biomass burning and
industrial activities. In contrast natural sources of WSON include mineral dust, bacteria, sea
salt, organic debris, natural biomass burning (Neff et al, 2002; Cornell et al., 2003; Mace et
al., 2003a, b, c; Gilbert et al., 2005; Sorooshian et al., 2008; Altieri et al., 2016). Atmospheric
organic nitrogen can also be formed through chemical reactions. For example, reactions
between volatile organic compounds, $NO_x$ and ammonium sulfate aerosols may lead to the
formation of nitrogen-containing compounds (Surratt et al., 2008; Galloway et al., 2009; De
Haan et al., 2011; Yu et al., 2011). Furthermore, atmospheric organic nitrogen plays an
essential role in many global processes which may impact on the chemistry of the atmosphere
as well as climate and biogeochemical cycles. Similar to ammonium, organic nitrogen species



such as urea and amines have acid-neutralizing capacities (Ge et al., 2011). It has been shown
that nitrogen containing organic compounds nucleate cloud droplets and may contribute
considerably to the indirect aerosol effect (Twohy et al., 2005). Phytoplankton and bacteria
production in aquatic environments has been found to be stimulated by the addition of water-
soluble organic nitrogen (Timperly et al., 1985; Peierlt and Paerl, 1997; Seitzinger and
Sanders, 1999). The laboratory experiments performed by Seitzinger and Sanders (1999)
demonstrated production of coastal marine bacteria and phytoplankton which are stimulated
by the addition of water-soluble organic nitrogen, 45-75 % being Bioavailable. From the mid
1800s to 2000, as a result of anthropogenic activities, reactive nitrogen and reactive
anthropogenic organic nitrogen have increased by almost 10 and 4 fold, respectively, leading
to a significantly modified global nitrogen cycle. This in term has impacted upon the marine
nitrogen biogeochemical cycling (Galloway et al., 2002, 2008; Duce et al., 2008).
The Mediterranean Sea is characterized by oligotrophic surface waters with Low
Nutrient Low Chlorophyll (LNLC) regions. This has been attributed to mainly anti-estuarine
(reverse thermohaline) circulation (Hamad et al., 2005). The Eastern Mediterranean (25) has
higher molar N/P ratios than those observed in the Western Mediterranean (22) and the
Redfield ratio (Krom et al., 2004; Yılmaz and Tuğrul, et al., 1998). It has been proposed that
the primary productivity in the Eastern Mediterranean is phosphorous limited (Thingstad et al.,
2005). However it has also been suggested that the primary productivity and bacterial activity
in the Eastern Mediterranean is limited by nitrogen or co-limited by nitrogen and phosphorous
(Yücel, 2013; Yücel, 2017).  Very little research has focused on the importance of water-
soluble organic nitrogen inputs to marine productivity in the Eastern Mediterranean (Violaki
and Mihalopoulos, 2010; Violaki et al., 2010). Hence, the unique contributions of the current
study will be to (i) define the temporal variability of atmospheric water-soluble organic
nitrogen, (ii) assign the origin of the water-soluble organic nitrogen, (iii) assess the influence



of mineral dust on water-soluble organic nitrogen and (iv) enhance our knowledge of the
quantitative dry and wet deposition for water-soluble organic nitrogen and its possible
influence on marine productivity in the North Eastern Mediterranean.
These will be achieved by using the acquired data from the analyses for water soluble
inorganic and organic nitrogen species of a series of size fractionated aerosol (coarse and fine)
and rain samples collected from March 2014 to April 2015 from the northern coast (Erdemli,
Turkey) of the Levantine Basin, Eastern Mediterranean.

**2. Material and Methods**
**2.1. Sampling Site Description**
Aerosol and rain sampling were carried out at a rural site located on the coast of the
Eastern Mediterranean, Erdemli, Turkey (36̍ 33′ 54″ N and 34̍ 15′ 18″ E). The sampling tower
(above sea level ~ 22 m, ~ 10 m away from the sea) is situated at the Institute of Marine
Sciences, Middle East Technical University (IMS-METU). Its immediate vicinity is
surrounded by cultivated land to the north and to the south of the Northern Levantine Basin.
Although the site is not under the direct influence of any industrial activities (soda and
fertilizer), the city of Mersin with a population of 800.000 is located 45 km to the east of the
sampling site (Kubilay and Saydam, 1995; Koçak et al., 2012) and hence aerosol and
rainwater samples influenced by air mass transport from the east may have been influenced by
these regional anthropogenic activities.

**2.2. Sample Collection and Preparation**
*Aerosol:* A Gent type stacked filter unit (SFU) was used to collect aerosol samples in two size
fraction (coarse: d = 10-2.5 μm and fine: d < 2.5 μm) (for more details see Hopke et al., 1997;
Koçak et al., 2007). Briefly, the first section of the filter holder was loaded with an 8 μm pore



size polycarbonate filter (Whatman Track Etched 111114, circle diameter: 47 mm), whilst the
second section was loaded with a 0.4 μm pore size polycarbonate filter (Whatman Track
Etched 111107, circle diameter: 47 mm). The cassette unit was then placed into the
cylindrical cassette holder, which is designed to prevent the intrusion of particles larger than
10 μm when the sampler is operated at a flow rate of 16.0-16.5 L/min. Daily (24 hours)
temporal sample resolution was carried out. Operational blank filters were processed in the
same way as the collected samples with the exception that no air was passed through the
filters. In order to minimize any possible contamination, the filter loading and unloading were
achieved in a laminar airflow cabinet.
The aerosol sampling campaign commenced in March 2014 and ended in April 2015.
During the sampling period, a total of 674 aerosol samples in two size fractions (coarse = 337;
fine = 337) were obtained. The observational coverage of the aerosol sampling period was 80
%. The sampling was terminated from time to time due to technical malfunction of the SFU
and/or cleaning procedure of sampling apparatus.

*Rain:* Rainwater samples were collected using an automatic Wet/Dry sampler (Model ARS
1000, MTX Italy). A total of 23 rain samples were collected during the sampling period. After
each rain event, the rainwater samples were immediately transferred to the laboratory for
filtration (0.4 μm Whatman, polycarbonate filters).

*Storage of Samples:* Aerosol and rainwater samples were stored frozen (-20 ˚C) immediately
after collection until analyses (not more than a month). Cape et al. (2001) have been shown
that there were no significant losses for inorganic and organic nitrogen during the storage
(freezing for 3 months) of rain samples with an added biocide.



*Sample Preparation:* In order to determine the concentrations of water-soluble nitrogen
species (WSTN, $NO_3^-$ and $NH_4^+$) and major water-soluble ions ($Cl^-$, $SO_4^{2-}$, $Na^+$, $K^+$, $Mg^{2+}$,
$Ca^{2+}$) in an aerosol sample, one quarter of the filter was extracted for 60 minutes in 20 mL of
ultra-pure water (18.2 Ωm) by mechanic shaking. About 100 μL chloroform (Merc 2444, 99.8
%) was added as a preservative to prevent biological activity after removing the filter
(Bardouki et al., 2003, Koçak et al., 2007). Before measuring the water-soluble species,
extracts were filtered with 0.4 μm pore size polycarbonate filters.

**2.3. Chemical Analysis**
*Water Soluble Total Nitrogen:* High Temperature catalytic oxidation (Torch Teledyne Tekmar
TOC/TN) was applied to determine the WSTN concentrations in the aerosol and rainwater
samples. The liquid aliquot of the sample is injected into the combustion furnace (750 $^{o}$C) and
the N in the sample was then converted to NO gas. The carrier gas (high purity dry air)
sweeps the sample into nondispersive infrared detector. From here, the sample is carried to
the nitrogen module.  In this unit NO is mixed with $O_3$ since the chemiluminescent detection
of NO is based on the reaction between NO and $O_3$. After the formation of excited nitrogen
dioxide ($NO_2^*$), the extra energy is given of as light when $NO_2^*$ relaxes to its ground state.
The light signal to an electronic signal for quantification is then measured by a
chemiluminescence detector with a photomultiplier tube.

*Water soluble Inorganic and Ancillary Species:* In addition to $NO_3^-$ and $NH_4^+$, major water-
soluble ions concentrations were measured by using a Dionex ICS-5000 Ion Chromatography
instrument. Water-soluble anions ($Cl^-$, $SO_4^{2-}$, $NO_3^-$) were determined by applying AS11-HC
separation column, KOH (30 mM) eluent and AERS-500 (4 mm) suppressor  whilst water-
soluble cations ($Na^+$, $K^+$, $Mg^{2+}$, $Ca^{2+}$)were detected electrochemically by using a CS12-A



separation column, MSA (20 mM) eluent and CSRS-300 (4 mm) suppressor (Product Manual
for Dionex IonPac AS11-HC-4m, IonPac CS12A Manual).

Blank values of WSTN for aerosol and rain samples were less than limit of detection

(20 ppb). The blank contributions of water-soluble ions in aerosol samples were found to be
less than 10 % and concentrations were corrected for blanks.

**2.4. Calculations**

WSON concentrations (see Eq. 1) were determined from the difference between the

individual concentrations of WSTN and water-soluble inorganic nitrogen (WSIN) (see Eq. 2)
since there is no direct analytical method to detect the concentration of water-soluble organic
nitrogen. The precision for WSON was calculated via using the formula (see Eq. 3) suggested
by Hansell (1993). The precision (75 nmol N m$^{-3}$) was found to be almost three times higher
(see Eq. 4, R ~ 0.3) thanthat of the arithmetic mean of WSON in aerosols whilst it (90 μmol N
L$^{-1}$) was estimated to be approximately four times larger than that of volume weighted mean
of WSON in rain. Such high values are not unusual. For example, if the data presented by
Mace et al. (2003a) would be used, precisions would have been 5 and 8 times higher
thanthose of the concentrations of WSON in aerosol and rain, respectively. Table 1 shows the
number of negative WSON values and the positive WSON biases for coarse and fine modes.
Correspondingly, about 5 (n=18) and 15 % (n=52) of the values were negative in coarse and
fine particles. The substitution with zero yielded 2 and 14 % positive bias for coarse and fine
mode whereas; the omission of zero resulted in 8 and 34 % positive bias in coarse and fine
WSON mean concentrations. Consequently, the presentation of the general characteristics of
the data includes all negative concentrations. In order to evaluate the variability in the aerosol
WSON and apply PMF, however, different approach was adopted. To this end, arbitrary
thresholds have been defined as the ratio between WSON mean concentration and the





calculated precision (see Eq. 4). Thus, during assessing the variability in aerosol WSON and
the application of PMF, WSON concentrations having R values larger than 0.3 will be
considered since the arbitrary threshold simply reduces the uncertainty.

$$WSON = WSTN - WSIN \ (1)$$

$$WSIN = NO_3^- + NH_4^+ \ (2)$$

$$S_{WSON} = (\ s_{WSTN}^2 + s_{WSIN}^2)^{1/2} \ (3)$$


$$R = \frac{WSON_{MEAN}}{S_{WSON}} \ (4)$$

The rain volume weighted average concentration ($C_w$) of nitrogen species can be
calculated as follow:

$$C_W = \frac{\sum_{i=1}^{n} C_i x Q_i}{\sum_{i=1}^{n} Q_i} \ (5)$$

The wet and dry atmospheric fluxes of nitrogen species were calculated according to
the procedure explained in Herut et al. (1999, 2002). The wet atmospheric deposition fluxes
($F_w$) were calculated from the annual precipitation ($P_{annual}$) and the volume weighted mean
concentration ($C_w$) of the substance of interest.

$$F_W = C_W \times P_{annual} \ (6)$$

The dry deposition ($F_d$) is calculated as the product of the atmospheric mean nutrient
concentrations ($C_d$) and their settling velocities ($V_d$), where $F_d$ is given in units of μmol m$^{-2}$
yr$^{-1}$, $C_d$ in units of μmol m$^{-3}$ and $V_d$ in units of m yr$^{-1}$.

$$F_d = C_d \times V_d \ (7)$$

The settling velocities ($V_d$, see Eq. 8) for each water-soluble nitrogen species were
calculated by using an approach adopted by Spokes et al. (2001). $C_c$ and $C_f$ refer to as the
relative contribution of coarse and fine modes and 2.0 and 0.1 cm s$^{-1}$ are deposition velocities
proposed by Duce et al. (1991) for coarse and fine particles respectively.



$$V_d = C_c \times 2.0 + C_f \times 0.1 \quad (8)$$


**2.5. Air Masses Back Trajectories and Airflow Classification**

Three day back trajectories of air masses at the four altitudes (1000, 2000, 3000 and
4000 meter) levels arriving at Erdemli station were computed by using the HYSPLIT
Dispersion Model (HybridSingle Particle Lagrangian Integrated Trajectory; Draxler and
Rolph, 2003). Three day back trajectories reaching at the altitude of 1000 m were classified
into six sectors: (i) Middle East, (ii) North Africa, (iii) Turkey, (iv) Eastern Europe, (v)
Western Europe and (vi) Mediterranean Sea in order to assess the influence of airflow on
WSON concentration in $PM_{10}$ (for more details see Koçak et al., 2012).

**2.6. Positive Matrix Factorization (PMF) for Source Apportionment of WSON**

The receptor modeling tool *Positive Matrix Factorization* (U.S. Environmental
Protection Agency PMF version 5.0, hereinafter referred to as 'PMF') was utilized to identify
the sources of WSON in $PM_{10}$ at Erdemli. PMF has been proven to be a robust tool in
characterizing the sources of aerosol (Paatero and Tapper, 1994; Huang et al., 1999; Lee et
al., 1999; Viana et al., 2008; Koçak et al., 2009). EPA PMF 5.0 software mainly consists of
Model Run and Rotational tools (see EPA/600/R-14/108; USEPA, 2014). Before application
of the software, the user must supply two input files namely, concentration and uncertainly.
The former contains concentrations of the aerosol species whilst the later has corresponding
uncertainty for each variable. Uncertainty was set to 5 % for each species with the exception
of WSON (15 %) since WSON exclusively donated high uncertainty. The base run of PMF
was achieved by setting the number of runs and random starting points (in other word seeds)
to 250 and 50, respectively. Base Model Displacement (DISP), Bootstrap (BS) and Bootstrap
Displacement (BS-DISP) methods were sequentially used after base run. The DISP accesses





the rotational ambiguity. DISP error estimates showed that there were no factor swaps and
significant decrease in Q during DISP, being 0 and 0.00, respectively. Therefore, DISP results
did not reveal rotational ambiguity, implying the solutions to be robust. Except in one case,
results from BS and BS-DISP (n=50) did not indicate any asymmetry and rotational
ambiguity for 5 factors. To evaluate the rotational ambiguity, different Fpeak values were
applied, considering changes in dQ to be less than 5 %. Furthermore, G-sape plots of Fpeak
solutions were examined to determine convergence toward the axis or lower/zero
contribution. Thus, Fpeak values of -0.7 was used and five factors were identified by using
PMF 5.0. BS of Fpeak at -0.7 did not reveal any swaps for five factors. The slope of the
estimated WSON against measured WSON was 10 % less than unity with correlation
coefficient and intercept of 0.87 and 1.5 (nmol N m$^{-3}$), respectively.

## 3. Results and Discussion

### 3.1. General Characteristics of the Data

In this section the general characteristics of the Water-Soluble Organic Nitrogen
(WSON), Nitrate ($NO_3^-$), Ammonium ($NH_4^+$) and Water-Soluble Total Nitrogen (WSTN) in
aerosol and rain will be discussed.

*Aerosol*: The statistical summary for WSON, $NO_3^-$, $NH_4^+$ and WSTN in $PM_{10}$ aerosol
samples obtained from Erdemli between March 2014 and April 2015 is presented in Table 2.
Among the nitrogen species, WSON exhibited the highest arithmetic mean, followed by
ammonium and nitrate concentrations respectively. The maximum concentration of WSON
was estimated to be 79 nmol N m$^{-3}$ with a mean value and standard deviation of 23.8 ± 16.3
nmol N m$^{-3}$. Approximately 66 % of the WSON was associated with coarse particles, the
remaining fraction (34 %) was present within the fine mode. A number of studies have



reported the relative size distribution of WSON for the Eastern Mediterranean marine aerosol
(Finokalia, Violaki and Mihalopoulos, 2010) and those observed at remote marine sites
(Hawaii, Cornell et al., 2001; Tasmania, Mace et al., 2003a). The aerosol WSON at Finokalia
(68 %) and Hawaii were primarily found in the fine mode whilst WSON in the south Pacific
marine aerosol (Tasmania) it was mainly associated with the coarse fraction. It is likely that
the WSON at Erdemli (a) is relatively less impacted by anthropogenic sources and/or (b) is
more influenced by mineral dust transport and resuspension of cultivated soil compared to
that observed at Finokalia.
$NO_3^-$ and $NH_4^+$ aerosol concentrations ranged between 0.2-88.4and 0.5-164.4 nmol N
m$^{-3}$ with mean values (standard deviations) of 17.9 (±15.7) and 23.3 (±24.4) nmol N m$^{-3}$. As
expected, $NO_3^-$ was mainly associated with coarse particles, accounting for 87 % of the
observed mean value whilst $NH_4^+$ was dominant in the by fine mode, contributing 96 % to
the detected mean concentration. Similar results have been reported for Eastern
Mediterranean marine aerosol (Bardouki et al, 2003; Koçak et al., 2007). The predominance
of $NO_3^-$ in the coarse mode might be due to gaseous nitric acid or other nitrogen oxides
reacting with alkaline sea salts and mineral dust particles. In contrast the occurrence of $NH_4^+$
in the fine fraction is mainly as a result of the reaction between gaseous alkaline ammonia and
acidic sulfuric acid (Mihalopoulos et al., 2007).
WSTN concentrations in aerosols varied between 9.7 and 176.5 nmol N m$^{-3}$ with
anarithmeticmean value of 63.5± 32.0 nmol N m$^{-3}$, respectively. The mean WSTN
concentration being almost equally influenced by coarse (51 %) and fine particles (49
%).Table 2 demonstrates the relative contributions of WSON, $NO_3^-$ and $NH_4^+$ to the WSTN in
PM$_{10}$. As can be deduced from the table, the WSTN concentration was equally influenced by
WSON and $NH_4^+$, each species contributing 37% and 35 %, respectively. In contrast the
contribution of $NO_3^-$ to WSTN was found to be 28 %.






**_Rain:_** Volume-weighted-mean (VWM) concentrations of WSON, $NO_3^-$, $NH_4^+$ and WSTN in
rainwater are presented in Table 2, along with the minimum and maximum concentrations as
well as the relative contributions of WSON, $NO_3^-$ and $NH_4^+$ to WSTN. As can be deduced
from table, VWM concentrations of each species were comparable, $NH_4^+$ exhibited the highest
concentration with a value of 28.7 μmol N $L^{-1}$. The VWM concentration of WSON and $NO_3^-$
were 21.5 and 23.3 μmol N $L^{-1}$, respectively. Considering their relative contributions to
WSTN, WSON and $NO_3^-$ account 29 and 32 % of the WSTN whilst $NH_4^+$ represented 39 % of
the observed WSTN concentration in rainwater.

**3.2. Comparison of WSON in Aerosol and Rain with data from the Literature**
The concentrations of WSON in marine aerosols and rain samples collected from
different sites located around the Mediterranean, Atlantic and Pacific regions are illustrated in
Table 3. Comparing the current WSON values with those reported in the literature is
challenging due to (i) different applied sampling periods, sampling and measurement
techniques and (ii) the high uncertainty associated with the estimation of WSON.
Furthermore, within the literature there is a lack of information defining the uncertainty of
WSON though there is a substantial statistical knowledge. Keene at al. (2002) in particular,
have highlighted the tendency in the literature to neglect negative values or substitute such
values with zero instead when calculating the WSON from the difference between WSTN
and WSIN. As these authors have highlighted the omission or substitution of such values
inevitably would result in a positive bias in the WSON concentrations.
In general the lowest concentrations in aerosols were found in those derived from
remote or pristine marine environments. The WSON concentrations in the atmosphere over
the Indian (Amsterdam Island: 1.0 nmol N $m^{-3}$, Violaki et al., 2015), Atlantic (Barbados: 1.3



nmol N m$^{-3}$, Zamora et al., 2011) and Pacific Ocean (Hawaii, Oahu: 4.1 nmol N m$^{-3}$, Cornell

et al., 2001, Tasmania: 5.3 nmol N m$^{-3}$, Mace et al., 2003b) were at least 4 times less than

those observed for Eastern Mediterranean (Erdemli: 23.8 nmol N m$^{-3}$, this study; Finokalia:

17.1 nmol N m$^{-3}$, Violaki and Mihalopoulos, 2010). These lower values might be attributed to

(i) the absence of the strong anthropogenic sources in the vicinity of the sampling sites and/or

(ii) the dilution of the WSON originating from long range transport via both dry and wet

deposition. The highest WSON concentrations emerged particularly over China (Ho et al.,

2015, concentration of WSON measured in PM$_{2.5}$) and Taiwan (Chen et al., 2010) with values

above 70 nmol N m$^{-3}$As stated in Chen et al. (2010) WSON concentrations at these sampling

sites were markedly influenced by anthropogenic activities such as fossil fuel combustion and

man induced biomass burning. Concentrations over the Amazon (Mace et al., 2003c) in the

dry season (61 nmol N m$^{-3}$) have also been noted. Such high values were ascribed to natural

fires (Mace et al., 2003c). The mean WSON concentration at Erdemli (23.8 nmol N m$^{-3}$) was

comparable to that reported previously for the same site (29 nmol N m$^{-3}$, Mace et al., 2003a).

In contrast, the present WSON concentration was almost 1.5 times higher than that observed

at Finokalia (Violaki and Mihalopoulos, 2010).

The reported WSON values for rain also exhibited the lowest concentrations in those

derived from remote or pristine marine environments, such as Hawaii (2.8 μmol N L$^{-1}$,

Cornell et al., 2001). The highest WSON concentrations were observed in China (North China

Plain: 103 μmol N L$^{-1}$, Zhang et al., 2008) and Norwich, UK (33 μmol N L$^{-1}$, Cornell et al.,

1998), respectively. These high values were again attributed to the anthropogenic sources.

### 3.3. Temporal Variability of Water-Soluble Nitrogen Species in Aerosol Erdemli

Fig.1 illustrates daily variation of the water-soluble nitrogen species in aerosol

samples together with the daily rainfall from March 2014 to April 2015. The same figure also



presents the concentrations in rainwater samples collected between October 2014 and April
2015. It is clear that WSON concentrations exhibited large variations from one day to another
day. The daily variability in the concentration of WSON may be an order of magnitude. Such
variability has also been reported in the Atlantic (Zamora et al., 2011), Pacific (Chen et al.,
2010) and Eastern Mediterranean marine aerosols (Violaki and Mihalopoulos, 2010). These
studies demonstrated that the daily change in the concentrations of WSON  arises from a
combination of (a) meteorological parameters (such as rain, temperature and wind
speed/direction), (b) chemical reactions, (c) history of air masses back trajectories and (d)
source emission strength.

In general, lower concentrations of WSON were found to be associated with rainy

days. To serve as an illustration, one of the lowest WSON concentrations was observed on
19$^{th}$ of October 2014, after two consecutive days of rainfall, with a value of 6 nmol N m$^{-3}$. In
contrast, one of the highest observed WSON concentrations (66.1 nmol N m$^{-3}$) was detected
on 2$^{nd}$ of March 2014 when the air mass back trajectories were associated with south/south
westerly airflow (for more details see section 3.4). Another high concentration of WSON was
observed on 5$^{th}$ of July 2014, with a value 66 nmol N m$^{-3}$.  94% of the WSON  was present in
the coarse mode, however, during this event there was no intense dust intrusion either from
the Sahara or from the Middle Eastern deserts. Corresponding OMI-AI index and nssCa$^{2+}$ (33
nmol m$^{-3}$) also supports this observation (see Fig.2). Lower layer air mass back trajectories
(1000 and 2000 m) demonstrated that Erdemli was under the influence of north/north westerly
airflow from Turkey after passing over Turkey's largest cultivated plain, Konya. Thus, this
high value might be attributed to re-suspension of the soil affected by intense agricultural
activities. On 20$^{th}$ of January 2016 the WSON concentration was 60 nmol N m$^{-3}$, 72 % being
present in the fine mode. For this event, the NH$_4^+$ concentration was 20 nmol N m$^{-3}$, two
times higher than the observed arithmetic mean in winter. Corresponding trajectories, AOD



(Aerosol Optical Depth) and AC (Angstrom Component) images are presented in Fig.3.
Airflow at 1 km showed air mass flow arriving at the sampling site from Turkey. AOD values
over the sampling site and coastline of Northeastern Mediterranean ranged from 0.2 to 0.5
whilst AC values demonstrated that the region was dominated by fine particles. Based on
above indicators, it may be concluded that anthropogenic sources were dominant.
A summary of the statistical analyses of the seasonal dataset of aerosol associated
WSON, $NO_3^-$ and $NH_4^+$ are shown in Table 4. The Mann-Whitney U test indicated that there
was a statistically significant difference among seasons, such that Summer > Spring ≈ Winter
> Fall. The arithmetic mean value of WSON in the summer was found to be 1.3 and 2.0 times
greater those observed for Spring/Winter and Fall, respectively. Furthermore, WSON was
chiefly associated with coarse particles in summer, amounting to in excess of 80%. This high
value in summer might be due to the absence of rainfall (see Fig.1) and enhanced re-
suspension of cultivated soil in the region. In summer, the mean concentration of $NH_4^+$ was
almost 2.4 times larger than all other seasons. The mean water-soluble $NO_3^-$ in summer was
1.4 high than that of spring. High $NH_4^+$ and $NO_3^-$ concentrations in summer might be
attributed again to the absence of rainfall and increase in incoming radiation. Similar results
have been reported for the Eastern Mediterranean (Bardouki et al., 2003).

**3.4. Influence of Mineral Dust Episodes on WSON aerosol concentrations**

As it is well documented, the Eastern Mediterranean Sea is heavily impacted by
mineral dust episodes originating from Sahara and the Middle East deserts (Kubilay and
Saydam, 1995; Kubilay et al., 2000, Koçak et al., 2004a, b and 2012).

For the current study between March 2014 and April 2015, water-soluble non-sea salt calcium
concentrations higher than 50 nmol m$^{-3}$ (2000 ng m$^{-3}$, as a threshold value) were defined as



mineral "dust events". These events were additionally confirmed using air mass back
trajectories and OMI-AI.  However, itt is worth mentioning that for samples containing
concentrations of nssCa$^{2+}$ less than 50 nmol m$^{-3}$ mineral dust transport from Sahara and the
Middle East deserts to sampling site may not be excluded, peculiarly in winter. Yet, the
application of such an arbitrary value is inevitable since it provides simplicity to explore if
there is any influence of mineral dust intrusion on WSON.

For example, one of the highest WSON concentrations (66.1 nmol N m$^{-3}$) was

observed  on 2$^{nd}$ of March 2014 when the air mass back trajectories was associated with
south/south westerly airflow. During this event, nssCa$^{2+}$ and NO$_3^-$ showed a dramatic increase
in their concentrations compared to those observed during the previous day, reaching up to
429 and 60 nmol m$^{-3}$, respectively.  OMI (Ozone Mapping Instrument) Aerosol Index (AI)
and three-day backward trajectory (1, 2, 3 and 4 km altitudes) air masses arriving at the
Erdemli sampling site on 2$^{nd}$ of March 2014 is shown in Fig.4. As can be seen from the figure,
all air masses (except at 1 km altitude) originated from North Africa whereas the back
trajectory for 1 km altitude exhibited airflow from the Middle East. Hence, suggesting that the
sampling site was under the influence of mineral dust transport originating from deserts
regions located at the Middle East and North Africa.. In support, the OMI-AI diagram clearly
indicates a large dust plume over the Eastern Mediterranean between coordinates 20-45 ˚N
and 15-40 ˚E. The Aerosol Index was found to be very high over the Northeastern
Mediterranean, ranging from 2.0 to 4.5.  During this dust episode, 85% of the WSON was
associated with the coarse fraction, which further supports mineral dust being  a main source
of water-soluble organic nitrogen.

Arithmetic mean concentrations together with corresponding standard deviations of

WSON, NO$_3^-$, NH$_4^+$ and nssCa$^{2+}$ for dust and non-dust events are presented in Fig.5. As can
be deduced from diagram, (except for NH$_4^+$,), WSON, NO$_3^-$ and nssCa$^{2+}$ indicated distinct



difference between dust and non-dust events. Indeed, the application of the non-parametric
Mann Whitney U test indicated statistically significant differences between dust and non-dust
events for WSON (p <0.03), $NO_3^-$ (p < 0.00002) and $nssCa^{2+}$ (p < 0.000001) whereas no
statistically significant difference were observed for $NH_4^+$,(p=0.56). The crustally derived
$nssCa^{2+}$ and anthropically derived $NO_3^-$ for dust events had arithmetic mean of 95.8 nmol m$^{-3}$
and 26.1 nmol N m$^{-3}$ which were almost four and two times higher than those of observed for
non-dust events, respectively. Such an increase in concentrations during dust events for these
species has been previously reported in the Eastern Mediterranean (Koçak et al., 2004b).
Similarly, the arithmetic mean of WSON (38.2 nmol m$^{-3}$) during dust events was 1.3 times
higher compared to the value observed during non-dust events (29.4 nmol m$^{-3}$). A similar
enrichment of WSON during dust events has been reported for Erdemli (Mace et al., 2003a;
Yellow Sea (Shi et al., 210) and Finokalia (Violaki and Mihalopoulos, 2010). In addition,
Griffin et al. (2007) have demonstrated a significant difference between dust and non-dust
events for bacterial and fungal colony forming units at Erdemli, the former being much
greater.. Thus, it might be speculated that this enhancement during dust events can be due to
(a) mineral dust borne microorganisms and/or (b) interaction (e.g. adsorption, acid-base
reaction) between mineral dust and organic nitrogen compounds.

**3.5. Impact of Airflow on WSON**
Arithmetic mean concentrations together with corresponding standard deviations for water-
soluble nitrogen species and $nssCa^{2+}$ in aerosol samples according to categorized air mass
sectors (at 1 km) are presented in Table 5 WSON concentrations can be broadly categorized
in two classes namely (a) Middle East, North Africa, Turkey and (b) Eastern Europe, Western
Europe and Mediterranean Sea. WSON concentrations in the first group were found to be at
least 1.2 higher than those observed for the second group. The application of the   Mann-



Whitney U test indicated that  there was a statistically significant difference in the
concentrations of WSON between the following air mass categories: North
Africa/Turkey/Middle East > Eastern Europe/Western Europe/ Mediterranean Sea ($p < 0.05$)
The highest $NO_3^-$ concentrations were associated with airflow from North Africa and Turkey
with value of 18 and 15 nmol N m$^{-3}$, respectively, and there was a statistically significant
difference compared to the remaining air mass sectors ($p > 0.05$). The mean concentrations of
$NO_3^-$ for air masses derived from North Africa and Turkey was at least 1.3 times larger than
those calculated for the Middle East, Eastern Europe, Western Europe and Mediterranean Sea
air sectors ($p > 0.05$). $NH_4^+$ had the highest concentration under the influence of airflow
derived from Turkey. For this airflow, detected concentration was 1.5-2.4 times greater than
those calculated for other air masses sectors. The Mann-Whitney test showed that there was a
statistically significant difference in the $nssCa^{2+}$ concentrations. Arithmetic mean
concentrations of $nssCa^{2+}$ in the Middle East and North Africa were approximately 2 times
higher compared to the remaining air masses. As expected, these two airflows were primarily
influenced by crustal material due to sporadic dust events originating from deserts located in
North Africa and the Middle East.

**3.6. Source Apportionment for WSON in Aerosol**
A number of studies have discussed the possible sources of WSON in aerosol material
by applying either simple correlation analyses (Mace et al., 2003c; Violaki and Mihapoulos,
2010; Ho et al., 2015) or multivariate factor analysis (Chen and Chen, 2010), including PMF
(Chen et al., 2010). Usage of correlation analyses is useful when the number in sample-
populationsare limited however; large datasets are required in order to carry out PMF and FA.
Direct and indirect emissions of WSON from the sea surface have been demonstrated
(Miyakazi et al., 2011; Altieri et al., 2016). Previous studies in the Eastern Mediterranean,



have observed WSON to be associated with mineral dust (Mace et al., 2003a; Violaki and
Mihalopoulos, 2010). As stated by Mace et al. (2003a), WSON might either have originated
from mineral dust or carried by dust events owing to adsorption of gaseous organic nitrogen
compounds onto pre-existing particles. In addition, Violaki and Mihalapoulos (2010) have
shown fossil fuel and biomass burning as sources of WSON to the Eastern Mediterranean
atmosphere.
Fig.6 describes the potential sources of WSON by applying PMF 5.0. The
predominant two factors were chiefly found to be related with water-soluble inorganic
nitrogen species. The first factor had a high-loading for $NH_4^+$ with a value of 0.81and a
moderate loading of $SO_4^{2-}$ (0.45).  These species, would suggest the formation of secondary
aerosols. As expected, the factor contribution plot (not shown) indicated summer maximum,
demonstrating accumulation of these particles due to the absence of rain and enhanced gas-to-
particle formation under the prevailing conditions (high temperature and solar radiation). The
equivalent ratio of $NH_4^+$ and $SO_4^{2-}$ for this factor was 0.79, indicating $(NH_4)HSO_4$
formation(Koçak et al., 2007). The second factor explained 77 % of the $NO_3^-$ variation and
described 17 and 10 % of the $SO_4^{2-}$ and $NH_4^+$, variation, respectively. This group was also
associated with cations such as $Na^+$ (11 %), $K^+$ (7 %), $Mg^{2+}$ (22 %) and $Ca^{2+}$ (29 %), implying
reactions mainly between acidic nitrate and alkaline species.  The first and second factors
accounted for 20 and 22 % of the variability in WSON, respectively.  It might, therefore be
argued that the variability of  WSON in the first group  resulted from the reaction between
volatile organic N and ammonium sulfate aerosols whilst the variability of WSON explained
by the second factor was as a result of  the reaction between volatile organic compounds and
$NO_x$ and/or neutralization of acidic nitrate by alkaline nitrogen-containing compounds such as
urea and amine The third factor was heavily influenced by $Cl^-$ (0.8) and $Na^+$ (0.70) while
moderately impacted by $Mg^{2+}$ and $K^+$. This factor is likely due to sea salt formation.. The



forth factor was predominantly impacted by $Ca^{2+}$ and hence may be attributed to crustal
material. Crustal sources explained 10 % of the WSON variability. The final defined factor
had a moderate loading of WSON (EV = 0.43, explained 43 %) while it was affiliated with
$Na^+$ (0.15), $K^+$ (0.22) and $Mg^{2+}$ (0.24). The factor contribution diagram denoted highest
values in summer (not shown) and hence it can be attributed to re-suspension of the soil
particularly affected by intense agricultural activities.

**3.7. Atmospheric Depositions of N-Species and Implications Regarding Marine**

**Production**

485     The atmospheric dry (21.3 mmol N $m^{-2}$ $yr^{-1}$) and wet (36.7 mmol N $m^{-2}$ $yr^{-1}$)deposition fluxes of WSON, $NO_3^-$ and $NH_4^+$ and WSTN from March 2014 and April 2015

are demonstrated in Table 6. The atmospheric deposition of water-soluble nitrogen (57.8mmol
N $m^{-2}$ $yr^{-1}$)was chiefly originated from wet deposition (36.7 mmol N $m^{-2}$ $yr^{-1}$), amounting to
63 % of the total  atmospheric deposition. This difference might be attributed to the water-
soluble ammonium, for instance, the atmospheric depositions of $NH_4^+$ (15.6 mmol N $m^{-2}$ $yr^{-1}$)
was dominated by wet deposition, contributing 92 % of the total ammonium atmospheric flux.
Whereas, the atmospheric flux of WSON and $NO_3^-$ were more or less equally influenced by
both deposition modes. Corresponding WSON (9.8 mmol N $m^{-2}$ $yr^{-1}$) and $NO_3^-$(10.0 mmol N
$m^{-2}$ $yr^{-1}$) contributions to dry deposition were found to be 46 % and 48 % respectively. In
contrast, $NH_4^+$(1.3 mmol N $m^{-2}$ $yr^{-1}$)was only estimated to contribute 6 % of the total
deposition. Wet deposition of nitrogen was impacted by WSON (10.8 mmol N $m^{-2}$ $yr^{-1}$), $NO_3^-$
(11.7 mmol N $m^{-2}$ $yr^{-1}$), and $NH_4^+$ (14.3 mmol N $m^{-2}$ $yr^{-1}$)in the increasing order 29 % < 32 %
< 39 %. On average, WSON accounted for 36 % of the total atmospheric deposition of
WSTN.





If one assumes that all N species are bioavailable to primary producers for primary

production and if a Redfield N/C ratio of 106/16 is applied, it would be estimated that

atmospheric N depositions can support new production 7.95 g C $yr^{-1}$. It has been shown that

annual primary production for coastal and open waters of Cilician Basin were around 413 mg

C $m^{-2}$ $d^{-1}$ and 179 mg C $m^{-2}$ $d^{-1}$, respectively (Yücel, 2013). It has been noted that f- ratio

(ratio between new and total production) may ranged between 0.05 and 0.16 in oligotropic

seas such as Mediterranean (Estrada, 1996 and references therein). If the f-ratio of 0.16 is

applied, the annual new production for coastal and open waters of Cilician Basin would be

24.15 g C $yr^{-1}$ and 10.5 g C $yr^{-1}$ respectively. Consequently, the atmospheric water-soluble

nitrogen flux was found to sustain 33 % of the new production in coastal and 76 % of it in

open waters.

**4. Conclusion**

In the current study, water-soluble organic nitrogen in aerosol and rain samples

obtained over the Eastern Mediterranean has been investigated. From this investigation the

following summary may be made:

1) Of the nitrogen species, aerosol WSON (23.8 ± 16.3 nmol N $m^{-3}$) exhibited the

highest arithmetic mean, followed by ammonium (23.3 ± 14.4 nmol N $m^{-3}$) and then nitrate

(17.9 ± 15.7 nmol N $m^{-3}$). Aerosol WSON was mainly associated with coarse particles (66

519       %). The WSTN was equally influenced by WSON and $NH_4^+$, each contributing 37 and 35 %,

respectively, whereas the contribution to WSTN of $NO_3^-$ was 28 %. In rainwater, the VWM

concentrations of water-soluble nitrogen species were comparable. WSON and $NO_3^-$

accounted for 29 and 32 % of the WSTN whilst $NH_4^+$ elucidated 39 % of the WSTN.

2) Aerosol WSON concentrations exhibited large variations from one day to another

524       day. Generally, lower concentrations were observed during rainy days. Higher concentrations


of aerosol WSON were associated with different airflow. The three highest concentrations
were related to (i) mineral dust transport from Sahara and the Middle East deserts, (ii)
north/north westerly airflow from Turkey's largest cultivated plain, Konya and (iii) mid-range
pollution transport from the Turkish coast.  3) Influence of mineral dust transport on aerosol
WSON concentrations was assessed. The crustally derived $nssCa^{2+}$ and anthropogenic $NO_3^-$
for dust events had arithmetic mean of 95.8 nmol m$^{-3}$ and 26.1 nmol N m$^{-3}$ which were almost
four and two times higher than those of observed for non-dust events. The arithmetic mean of
WSON (38.2 nmol m$^{-3}$) for dust events was 1.3 times higher compared to that observed for
non-dust events (29.4 nmol m$^{-3}$).

4) Source apportionment suggested that aerosol WSON was mainly originated from

anthropogenic sources including agricultural (43 %), secondary aerosols (20 %) and nitrate
(22%), whereas, the two natural sources crustal material (10 %) and sea salts (5%) contributed
15 % to the WSON.

5) The total atmospheric deposition of water-soluble nitrogen (57.8 mmol N m$^{-2}$ yr$^{-}$

$^1$) was mainly via wet deposition (36.7 mmol N m$^{-2}$ yr$^{-1}$). In contrast the atmospheric fluxes of
WSON and $NO_3^-$ were equally influenced by the dry and wet deposition modes. On average,
WSON accounted for 36 % of the total atmospheric deposition of WSTN. The annual new
production for coastal and open waters of Cilician Basin was estimated to be 24.15 g C yr$^{-1}$
and 10.5 g C yr$^{-1}$, respectively.  Using these estimates the atmospheric water-soluble nitrogen
flux could sustain 33 % of the new production in coastal and 76 % of it in open waters.







The authors declare that they have no conflict of interest.
**Acknowledgments**
This work was mainly supported by The Scientific and Technological Research Council of
Turkey (TUBITAK). Required data were collected within the framework of the TUBITAK
113Y107 project. This study was also supported by the DEKOSIM (Center for Marine
Ecosystem and Climate Research) Project (BAP-08-11-DPT.2012K120880) funded by
Ministry of Development of Turkey. We would like to thank to Ersin Tursak, Pınar Kalegeri
and Merve Açıkyol for helping during sample collection and analysis. Aerosol optical
thickness, angstrom component and aerosol index values used in this study were produced
with the Giovanni online data system, developed and maintained by the NASA GES DISC.
We also acknowledge the MODIS and OMI mission scientists and associated NASA
personnel for the production of the data used in this research effort.

M. Koçak developed the concept and designed the study. M. Nehir and M. Koçak performed
the experiments, analyzed the data and prepared the manuscript.



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



**Figure Captions**

**Figure 1.** The daily variations in the concentrations of (a) WSON, (b) $NO_3^-$ and (c) $NH_4^+$ (nmol N $m^{-3}$) together with rain amount (mm) from March 2014 and April 2015 for PM10.

**Figure 2.** Three day back trajectories showing the transport of air masses 1000m (black circle), 2000m (black star), 3000m (black square) and 4000m (black triangle) on 5th of July 2014 for Erdemli. Aerosol Index (AI) from OMI (Ozone Mapping Instrument) distribution also illustrated with a color bar from grey to black.

**Figure 3.** Three day back trajectories showing the transport of air masses 1000m (black circle), 2000m (black star), 3000m (black square) and 4000m (black triangle) on 20th of February 2015 for Erdemli. Aerosol Optical Depth (AOD, a) and Angstrom Component (AC, b) from MODIS (Moderate Resolution Imaging Spectroradiometer) distribution also demonstrated with a color bar from grey to black.

**Figure 4.** Three day back trajectories indicating the transport of air masses 1000m (black circle), 2000m (black star), 3000m (black square) and 4000m (black triangle) on 2nd of March 2014 for Erdemli. Aerosol Index (AI) from OMI (Ozone Mapping Instrument) distribution also illustrated with a color bar from grey to black.

**Figure 5.** Arithmetic means together with corresponding Standard deviations of WSON, $NO_3^-$, $NH_4^+$ and $nssCa^{2+}$ for dust and non-dust events at Erdemli site. Dark and light grey bars denote arithmetic mean for dust and non-dust, respectively. Black vertical line shows standard deviation.

**Figure 6.** Source apportionment of WSON from Positive Matrix Factorization for PM10 at Erdemli.

**Table Captions**

**Table 1.** The number of negative WSON values and positive biases in coarse and fine particles at Erdemli.

**Table 2.** The statistical summary of the WSON, $NO_3^-$, $NH_4^+$ and WSTN for aerosol (nmol N $m^{-3}$) and rain (μmol N $L^{-1}$) samples collected at Erdemli from March 2014 to April 2015.

**Table 3.** Comparison of WSON concentrations in aerosol (nmol N $m^{-3}$) and rain (μmol N $L^{-1}$) samples for different sites of the World.

**Table 4.** Seasonal statistical summary of the WSON, $NO_3^-$, $NH_4^+$, WSTN (nmol N $m^{-3}$) and $nssCa^{2+}$ (nmol $m^{-3}$) in aerosol samples collected at Erdemli from March 2014 to April 2015.

**Table 5.** Arithmetic means along with standard deviations of WSON, $NO_3^-$, $NH_4^+$ (nmol N $m^{-3}$) and $nssCa^{2+}$ (nmol $m^{-3}$) in aerosol samples as a function of the classified airflow corresponding to three day air mass back trajectories reaching at Erdemli.

**Table 6.** Atmospheric dry and wet deposition of WSON, $NO_3^-$, $NH_4^+$ and WSTN together with their relative contributions at Erdemli during the period of March 2014 and April 2015.





**Figures**

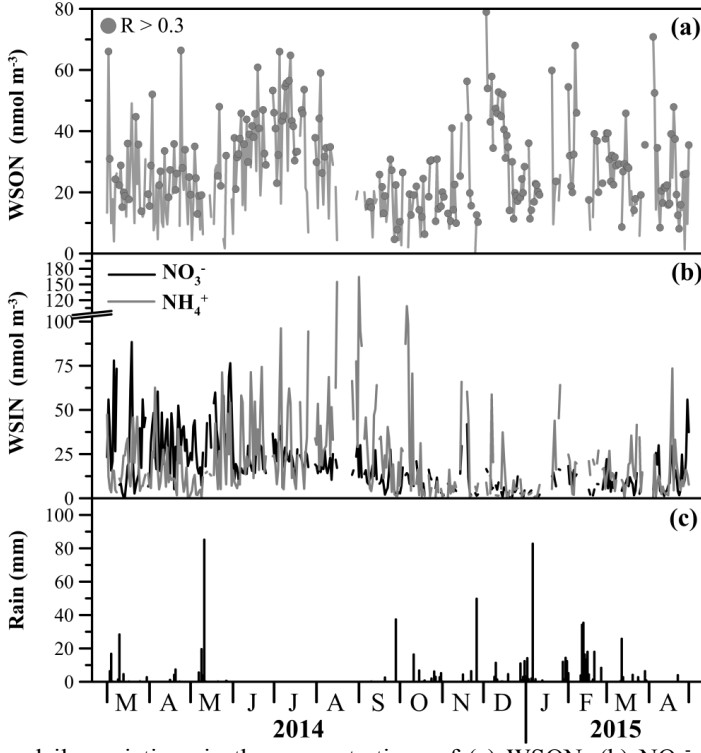

**Figure 1.** The daily variations in the concentrations of (a) WSON, (b) $NO_3^-$ and (c) $NH_4^+$
(nmol N m$^{-3}$) together with rain amount (mm) from March 2014 and April 2015 for $PM_{10}$.

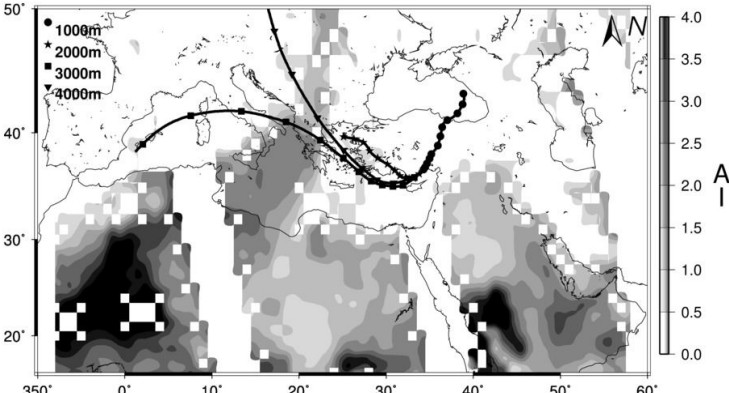

**Figure 2.** Three day back trajectories showing the transport of air masses 1000m (black
circle), 2000m (black star), 3000m (black square) and 4000m (black triangle) on 5$^{th}$ of July
2014 for Erdemli. Aerosol Index (AI) from OMI (Ozone Mapping Instrument) distribution
also illustrated with a color bar from grey to black.




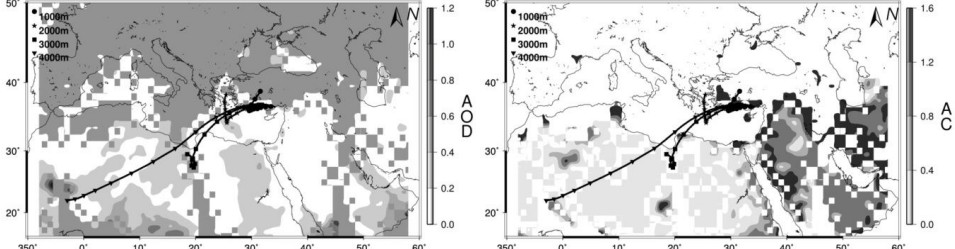

**Figure 3.** Three day back trajectories showing the transport of air masses 1000m (black
circle), 2000m (black star), 3000m (black square) and 4000m (black triangle) on 20[th] of
February 2015 for Erdemli. Aerosol Optical Depth (AOD, a) and Angstrom Component (AC,
b) from MODIS (Moderate Resolution Imaging Spectroradiometer) distribution also
demonstrated with a color bar from grey to black.

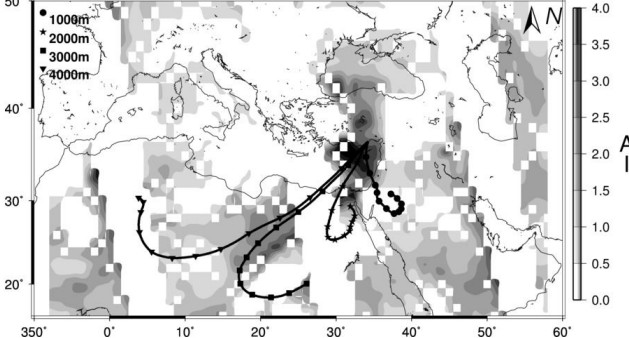

**Figure 4.** Three day back trajectories indicating the transport of air masses 1000m (black
circle), 2000m (black star), 3000m (black square) and 4000m (black triangle) on 2[nd] of March
2014 for Erdemli. Aerosol Index (AI) from OMI (Ozone Mapping Instrument) distribution
also illustrated with a color bar from grey to black.





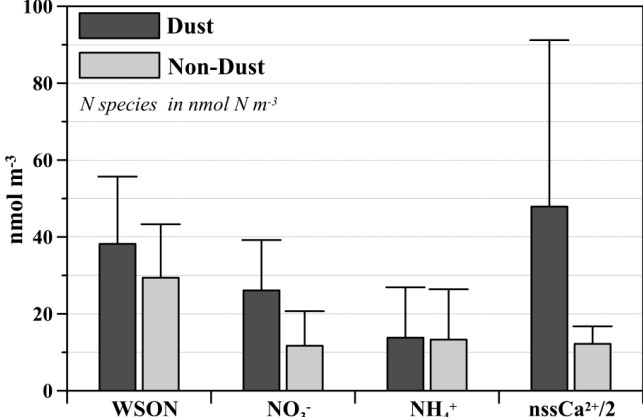

**Figure 5.** Arithmetic means together with corresponding standard deviations of WSON, $NO_3^-$, $NH_4^+$ and $nssCa^{2+}$ for dust and non-dust events at Erdemli site. Dark and light grey bars denote arithmetic mean for dust and non-dust, respectively. Black vertical line shows standard deviation.

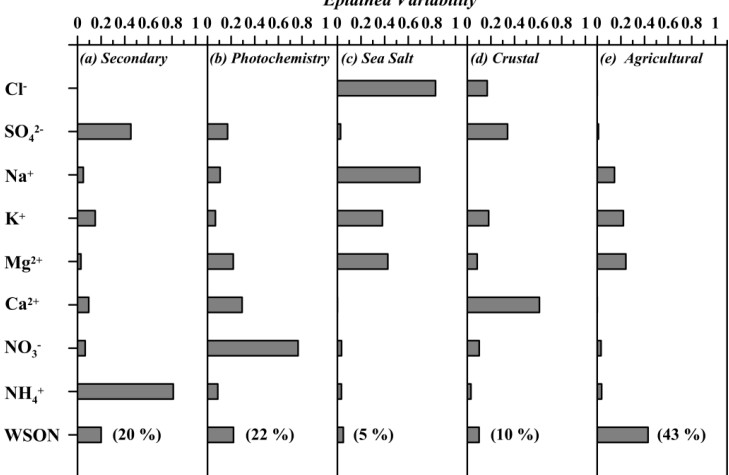

**Figure 6.** Source apportionment of WSON from Positive Matrix Factorization for $PM_{10}$ at Erdemli.



**Tables**

**Table 1.** The number of negative WSON values and positive biases in coarse and fine particles at Erdemli.

|  | Coarse | Fine |
|---|---|---|
| *Number of Samples* | 337 | 337 |
| *Number of Negatives* | 18 | 52 |
| $SZ^{1}$-*Positive Bias (%)* | 2 | 14 |
| $PZ^{2}$-*Positive Bias (%)* | 8 | 34 |

*1 and 2 refer to as the Substitution with Zero and the Omission of Zero, respectively.*

**Table 2.** The statistical summary of the WSON, $NO_3^-$, $NH_4^+$ and WSTN for aerosol (nmol N $m^{-3}$) and rain ($\mu$mol N $L^{-1}$) samples collected at Erdemli from March 2014 to April 2015.

| *AEROSOL (nmol N $m^{-3}$)* | WSTN | WSON | $NO_3^-$ | $NH_4^+$ |
|---|---|---|---|---|
| Arithmetic Mean | 63.5 | 23.8 | 17.8 | 21.9 |
| Standard Deviation | 32.0 | 16.3 | 15.2 | 23.8 |
| Minimum | 9.7 | -27.9 | 0.2 | 0.5 |
| Maximum | 176.5 | 79.0 | 88.4 | 164.4 |
| Coarse/$PM_{10}$ (%) | 51 | 66 | 87 | 4 |
| *Relative Contribution to WSTS (%)* |  | *37* | *28* | *35* |
| *RAIN ($\mu$mol N $m^{-3}$)* |  |  |  |  |
| VWM* | 73.5 | 21.5 | 23.3 | 28.7 |
| Minimum | 24.3 | -2.9 | 0.2 | 9.1 |
| Maximum | 356.2 | 257.2 | 74.6 | 122.6 |
| *Relative Contribution to WSTS (%)* |  | *29* | *32* | *39* |

*VWM refers to Volume Weighted Mean





857

**Table 3.** Comparison of WSON concentrations in aerosol (nmol N m$^{-3}$) and rain (µmol N L$^{-1}$)
samples for different sites of the World.

| Aerosol (nmol N m$^{-3}$) | WSON | NS | SP | Reference |
|---|---|---|---|---|
| **Mediterranean Sea** | | | | |
| *Erdemli, Turkey* | 23.8 | 674 | 2014-2015 | This Study |
| *Erdemli, Turkey* | 29 | 39 | 2000 | Mace et al. [2003a] |
| *Finokalia, Crete* | 17.1 | 65 | 2005-2006 | Violaki and Mihalopoulos [2010] |
| **Pacific Ocean** | | | | |
| *Hawaii* | 4.1 | 16 | 1998 | Cornell et al. [2001] |
| *Tasmania* | 5.3 | 24 | 2000 | Mace et al. [2003b] |
| *Taiwan* | 75.9 | 77 | 2006 | Chen et. al. [2010] |
| *Xi'an, China (PM$_{2.5}$)* | 300 | 65 | 2008-2009 | Ho et. al. [2015] |
| **Atlantic Ocean** | | | | |
| *Barbados* | 1.3 | 57 | 2007-2008 | Zamora et al. [2011] |
| *Amazon, dry season* | 61 | 37 | 1999 | Mace et al. [2003c] |
| *Amazon, wet season* | 3.5 | 27 | 1999 | Mace et al. [2003c] |
| **Indian Ocean** | | | | |
| *Amsterdam Island* | 1 | 42 | 2005 | Violaki et al. [2015] |
| Rainwater (µmol N L$^{-1}$) | WSON | NS | SP | Reference |
| **Mediterranean Sea** | | | | |
| *Erdemli, Turkey* | 21.5 | 23 | 2014-2015 | This Study |
| *Erdemli, Turkey* | 15 | 18 | 2000 | Mace et al. [2003a] |
| *Finokalia, Crete* | 18 | 18 | 2003-2006 | Violaki et al. [2010] |
| **Pacific Ocean** | | | | |
| *Tahiti\** | 4.8 | 8 | | Cornell et al. [1998] |
| *Hawaii* | 2.8 | 17 | 1998 | Cornell et al. [2001] |
| *Tasmania* | 7.2 | 6 | | Mace et al. [2003b] |
| *North China Plain, China* | 103 | 15 | 2003-2005 | Zhang et al. [2008] |
| *Kilauea, Hawaii* | 6.5 | 20 | 1998 | Cornell et al. [2001] |
| **Atlantic Ocean** | | | | |
| *Bermuda* | 5.6 | 5 | 1994 | Cornell et al. [1998] |
| *Mace Head* | 3.3 | 7 | | Cornell et al. [1998] |
| *Norwich, UK* | 33 | 12 | | Cornell et al. [1998] |
| *Virginia, US* | 3.1 | 83 | 1996-1999 | Keene et al. [2002] |
| *Delaware, US* | 4.2 | 50 | 1997-1999 | Keene et al. [2002] |
| *New Hampshire, US* | 0.6 | 12 | 1997 | Keene et al. [2002] |

*RC, NS and SP refer to relative contribution of WSON to WSTN, number of samples and sampling period,
respectively.*

**Table 4.** Seasonal statistical summary of the WSON, NO$_3^-$, NH$_4^+$, WSTN (nmol N m$^{-3}$) and
nssCa$^{2+}$ (nmol m$^{-3}$) in aerosol samples collected at Erdemli from March 2014 to April 2015.

| Aerosol *Species* | Winter | Spring | Summer | Fall |
|---|---|---|---|---|
| WSON | 33±16 | 28±13 | 41±11 | 20±10 |
| NO$_3^-$ | 7±5 | 15±12 | 21±7 | 9±8 |
| NH$_4^+$ | 10±12 | 11±9 | 24±16 | 10±13 |
| nssCa$^{2+}$ | 28±13 | 28±13 | 28±13 | 41±11 |
| Number of Samples | 47 | 79 | 46 | 44 |
| **Meteorology** *Parameter* | **Winter** | **Spring** | **Summer** | **Fall** |
| T ($^o$C) | 11±3 | 16±3 | 27±12 | 20±15 |
| Rain (mm) | 78 | 118 | 0.5 | 132 |
| Number of Rain Events | 16 | 16 | 2 | 15 |





**Table 5.** Arithmetic means along with standard deviations of WSON, $NO_3^-$, $NH_4^+$ (nmol N $m^{-3}$) and $nssCa^{2+}$ (nmol $m^{-3}$) in aerosol samples as a function of the classified airflow corresponding to three day air mass back trajectories reaching at Erdemli

| *Airflow* | WSON | $NO_3^-$ | $NH_4^+$ | $nssCa^{2+}$ |
|---|---|---|---|---|
| *Middle East* | 33±12 | 12±12 | 13±15 | 48±71 |
| *North Africa* | 36±16 | 18±11 | 12±14 | 46±38 |
| *Turkey* | 32±13 | 15±10 | 19±15 | 23±9 |
| *Eastern Europe* | 26±14 | 10±9 | 10±8 | 21±9 |
| *Western Europe* | 26±14 | 10±8 | 11±9 | 20±7 |
| *Mediterranean Sea* | 22±10 | 10±8 | 8±6 | 19±8 |

**Table 6.** Atmospheric dry and wet deposition of WSON, $NO_3^-$, $NH_4^+$ and WSTN together with their relative contributions at Erdemli during the period of March 2014 and April 2015.

| Species | $F_d$ (mmol N $m^{-2}$ $yr^{-1}$) | Relative Contribution |
|---|---|---|
| WSON | 9.8 | 46 |
| $NO_3^-$ | 10.0 | 48 |
| $NH_4^+$ | 1.3 | 6 |
| WSTN | 21.1 | |
| **Species** | **$F_w$ (mmol N $m^{-2}$ $yr^{-1}$)** | **Relative Contribution** |
| WSON | 10.7 | 29 |
| $NO_3^-$ | 11.7 | 32 |
| $NH_4^+$ | 14.3 | 39 |
| WSTN | 36.7 | |