# Peer review of "Atmospheric Water-Soluble Organic Nitrogen (WSON) in the Eastern Mediterranean: Origin and Ramifications Regarding Marine Productivity"

_Atmospheric Chemistry and Physics, 2017_

## Referee Comment (RC1) · Anonymous Referee #1 · 25 Aug 2017

Review of Nehir and Koçak This paper reports analyses of a very large set of aerosols and a small set of rainwater samples for major ions and water soluble organic nitrogen (WSON) from the Turkish sampling site on the Mediterranean coast. There have been similar studies at this site and at neighbouring sites over recent years, but the very large size of this data set makes this data set useful. The analyses seem to generally have been well done and the interpretation is quite thorough, but there are some parts of the paper that I think could be improved for final publication as described below. Introduction There have been some recent reviews of WSON which the authors might

reference (e.g. Cape et al., 2011 Atmos. Res 102, 30-48) since they summaries much of the material and offer a somewhat wider perspectives and more recent information on the composition of WSON. There is also now a global model of WSON (Kanakidou et al., 2012 Global Biogeochem. Cycl. 26, doi 10.1029/2011GB004277) which has contributed to an updated global nitrogen cycle revising the Duce et al 2008 paper cited (Jickells et al.;, 2017 Global Biogeochem. Cycl. 31, doi 10.1029/2016GB005586). Line 60, while amines will neutralise acids, it is not obvious the rest of WSON will. Line 62 I don't think Twohy discusses WSON Line 80 and later on, there is really pretty clear evidence that the Eastern Mediterranean is P limited. There is a vast body of work by Krom and colleagues that supports this (see most recently Pawley et al 2017 Global Biogeochem. Cycl. 31, 1010-1031 and the earlier summary in Krom et al 2010 Prog. in Oceanography 85, 236-244) and my reading of the Yücel 2017 paper does not actually contradict this view. Line 81-2 It is mentioned a little bit later on, but not here, that Mace et al have reported WSON from exactly the same site as the study here. This should be noted here and also in section 3.1. Note also the reference list lacks dates and while in the text the authors refer to Mace et al a,b and c, these are not identified in the references by these letters. Analytical Methods. In general the results seem to be of good quality, although there is no mention of how blanks were determined (i.e. what procedures were used to create blank samples for analysis), what standards were used in analysis and whether any certified reference materials were used. I do not really understand what the sentence line 163-4 about blanks being <10% means, is this true for all ions?. On line 163 20ppb is ambiguous, is it as ppb nitrogen and why not use molar units as elsewhere in the paper? Section 2.4 discusses the quite well known challenges of estimating WSON and its relatively low precision as a derived quantity (see Cape et al for instance). The precision of WSON depends a lot on the relative concentrations of the three components of the total nitrogen analysis, so it is not possible really to quote a single number. The authors discussion e.g. lines 170-174 and 175 (and lines 221-222) does not really explain what they actually estimate the precision to be. The use of PMF (which I am no expert on) here seems to require

provision of precision estimates, but I do not understand how the arbitrary thresholds used here (line 185-7) were arrived at or how sensitive the results are to these values. Section 2.6 As I understand it PMF is a form of principal component analysis and hence is an appropriate tool for this kind of source apportionment. I would suggest the authors may be better putting an explanation of the principal of the method here and putting the highly technical discussion into some sort of appendix, because I think many readers will not really be able to follow this section. Section 3.1 and 3.2 I wonder if these sections could be shortened a bit given that the results are broadly in line with other work in this region Line 304-307 I do not disagree with the interpretation here, but it is worth noting that this does carry the implicit assumption that land based sources dominate the emission of WSON. Section 3.3. This section is very general and the issue is approached in a more quantitative manner in 3.5 and 3.6, so I wonder if the section could be shortened. Section 3.4 Mace et al suggested that the Saharan dust was a major source of WSON at this site and they did this I think by a correlation between nssCa2+ and WSON. Here the association with dust seems to be weaker but the discussion does not really address this point, but simply notes there is an association with dust. This could be discussed further. Section 3.5 In Table 5 the WSON and other parameters are classified into 5 groups, but in the text here the discussion splits the data into two. It would be easier for the reader if the manuscript discussion and the tables did one or other of these, rather than mix them up in this way. Section 3.6 As noted earlier I am no expert on PMF. The striking thing for me from Figure 6 and the discussion, is that WSON does not resolve in any simple way into any of the components identified, emphasising the multiplicity of sources that it has, and this is particularly striking within such a large data set. I would also query the interpretation of what the associations mean (lines 469-474). The authors interpret the results in terms of formation mechanisms, but an alternative explanation might be emission sources. Section 3.7 As noted earlier the Eastern Mediterranean appears to be phosphorus limited. If this is the case then the addition of nitrogen will not necessarily stimulate any additional primary production, but rather contribute to the high N/P ratio (see earlier

[Figure]

Krom and Pawley references) and so the hypothesis behind the calculation (line 500-509) is flawed and the conclusions about the impacts on new production are incorrect. Section 4. This is really a summary and not a conclusion and simply repeats the earlier material.

---

## Referee Comment (RC2) · Anonymous Referee #2 · 28 Nov 2017

Review of the article entitled "Atmospheric Water-Soluble Organic Nitrogen (WSON) in the Eastern Mediterranean: Origin and Ramifications Regarding Marine Productivity" by Münevver Nehir and Mustafa Koçak.

An interesting article dealing with occurrence, size distribution and impact of WSON, a class of compounds not well studied up to date. The manuscript is well written and deserves publication once the authors addressing the following points. Abstract: - Aerosol WSON concentrations exhibited large temporal variations mainly due to rain and the origin of air mass flow. Rain scavenges all compounds why preferably WSON?

[Figure]

the authors should be more clear, e.g better say meteorology instead on rain -Line 27: agricultural activities (43 %), secondary aerosol (20 %), nitrate (22 %), crustal (10 %) and sea-salt (5 %). NO3 is a secondary aerosol? Please be more specific. See also comments below regarding sources attribution. -Line 29: Considering the Cilician Basin, the atmospheric water soluble nitrogen flux would sustain 33 % and 76 % of the new production in the associated coastal and open waters, respectively. Cilisian Bassin as part of the E. Mediterranean is mainly P limited. Thus explaining productivity on the basis of N only could lead to erroneous results. I suggest to remove this part or to be more cautious. Experimental section. • Line 122: The observational coverage of the aerosol sampling period was 80. Did the samples were uniformly distributed all over seasons? • Line 164: Did the authors perform recovery experiments with well-known mixtures of organic and inorganic compounds? It is reported that High Temperature catalytic oxidation is not fully recovering WSON from mixtures with inorganic substances. Results: • Please also report median in addition to average. Also how mean and median values changes if all data are considered (sensitivity test). • What is the seasonality of coarse/fine ratio of WSON? Also how this ratio varies as a function of air masses origin or better in dust vs non-dust samples? • At Table 4, the number of samples reported is only 216, whereas at line 120 they report analysis of 337 results. Why this difference? They correspond to samples with precision larger than 0,3? If yes what is the seasonal representativity of these 216 samples? • In rain water how many values have been omitted with precision lower than 0,3? What is the temporal variability of the omitted data?

Sources attribution (PMF). Using only IC data, source attribution of WSON is highly subjective. No ancillary data such as metals or OC/EC, NOx are available? Why no NH4 is found in agricultural factor? Given the compounds associated with this factor better assign it to soil re-suspension. Also factor A should better attributed to long-range transport (regional sources) due to the presence of NH4 and SO4. Similarly factor B with the presence mainly of NO3 could be better attributed to Anthropogenic sources (combustion) Line 487, better replace water-soluble nitrogen by WSTN.

Finally as said before, I strongly disagree with the utility of paragraph 3.7. The Eastern Mediterranean area is P limited thus all primary marine calculations should be based on P availability or better P and N availability. The conclusions presented at 3.7 are thus questionable.
* * *

---

## Author Comment (AC2) · 5 Jan 2018

Q1: Aerosol WSON concentrations exhibited large temporal variations mainly due to rain and the origin of air mass flow. Rain scavenges all compounds why preferably WSON?

A1: I agree. Rain scavenges all compounds. Meteorology was used instead of rain (see page 1, line 21).

Q2: NO3 is a secondary aerosol?

[Figure]

A2: NO3 is a secondary aerosol (please see A10).

Q3: Considering the Cilician Basin, the atmospheric water soluble nitrogen flux would sustain 33% and 76% of the new production in the associated coastal and open waters, respectively. I suggest to remove this part or to be more cautious.

A3: The sentence (the last paragraph of the section 3.7) was removed as suggested.

Q4: Did the samples were uniformly distributed all over seasons?

A4: Information about the seasonal observational coverage was added to the experimental section (see answer to question 8).

Q5: Did the authors perform recovery experiments with well-known mixtures of organic and inorganic compounds?

A5: Recovery experiments were carried out by using nitrate, ammonium, urea and mixture of all three species. Furthermore, the performance of our laboratory was studied by participating Quality Assurance of Information for Marine Environmental Monitoring in Europe (QUASIMEME) program. More information about recovery experiments was supplemented (see lines from 169 to 179).

Q6: Please also report median in addition to average. Also how mean and median values changes if all data are considered (sensitivity test).

A6: Median values were typed in Table 2. Number of samples were also included. Table 2 shows statistical summary for all samples (see line 287 and 288). Median values for WSON, NO3-, NH4+ and WSTN were respectively 10 %, 20 %, 40 % and 10 % lower than those that of arithmetic means.

Q7: What is the seasonality of coarse/fine ratio of WSON? Also how this ratio varies as a function of air masses origin or better in dust vs non-dust samples?

A7: Details about coarse/fine ratio of WSON for each season, air mass and dust/non-dust were given in Table 4 and 5 (also see lines 408-410, 458-460 and 477-480).

Q8: At Table 4, the number of samples reported is only 216, whereas at line 120 they report analysis of 337 results. Why this difference? They correspond to samples with precision larger than 0,3? If yes what is the seasonal representativity of these 216 samples?

A8: A total of 216 aerosols corresponded to samples with precision larger than 0.3. The observational coverage for winter, spring, summer and fall was respectively 60 %, 92 %, 81 % and 79 %. The seasonal observational coverage, after applying precision value of 0.3 (for more details see section 2.4 and Eq.4), was found to be comparable for winter (49 %), spring (53 %), summer (51 %) and fall (52 %) (see lines 129-132).

Q9: In rain water how many values have been omitted with precision lower than 0,3? What is the temporal variability of the omitted data?

A9: During the calculations of the dry and wet deposition, the values presented in Table 1 were utilized. Therefore, none of the aerosol and rain samples was omitted (see lines 208-209 and 221-222).

Q10: Sources attribution (PMF). Using only IC data, source attribution of WSON is highly subjective. No ancillary data such as metals or OC/EC, Nox are available? Why no NH4 is founding agricultural factor? Given the compounds associated with this factor better assign it to soil re-suspension. Also factor A should better attributed to long-range transport (regional sources) due to the presence of NH4 and SO4. Similarly factor B with the presence mainly of NO3 could be better attributed to anthropogenic sources (combustion) Line 487, better replace water-soluble nitrogen by WSTN.

A10: I agree that usage of ancillary data such metals and OC/EC during PMF would yield better results. Unfortunately there was no ancillary data such as metals, OC/EC and NOx. However, factor 1, 2, 3 and 4 were comparable to those obtained for Erdemli by using water-soluble ions and metals (Koçak et al., 2009). Factors 1, 2 and 5 were respectively attributed to ammonium-bisulfate (regional), nitrate (combustion) and soil re-suspension as suggested. Small amount of NH4 (3.7 %) was associated with factor

5. As stated in section 3.1, 96 % of the NH4 was originated from fine mode. Moreover, considering the first 20 % of the highest loadings in factor 5 (re-suspension), there was strong relationship between WSON and NH4 (r = 0.64). Thus, it sees that NH4 was reasonably associated with re-suspension. Water-soluble nitrogen was replaced by WSTN (see 515-518, 522-524 and Figure 6).

Please also note the supplement to this comment:
https://www.atmos-chem-phys-discuss.net/acp-2017-601/acp-2017-601-AC2-supplement.pdf

**Supplement:**

[revised manuscript text omitted]

The Mediterranean Sea is characterized by oligotrophic surface waters with Low Nutrient Low Chlorophyll (LNLC) regions. This has been attributed to mainly anti-estuarine (reverse thermohaline) circulation (Hamad et al., 2005). The Eastern Mediterranean (25) has higher molar N/P ratios than those observed in the Western Mediterranean (22) and the Redfield ratio (Krom et al., 2004; Yılmaz and Tuğrul, et al., 1998). Generally, the primary productivity in the Eastern Mediterranean is phosphorous limited (Krom et al., 1991; Krom et al., 2010; Powley et al., 2017). Depending on season, the limitation by nitrogen or co-limitation by nitrogen and phosphorus in the Eastern Mediterranean have been reported (Yücel, 2013; Yücel, 2017 and references therein). Based on molar N/P ratios in the atmospheric input (order of magnitude higher than that of Redfield, Markaki et al., 2003, 2010 ; Koçak et al., 2010) and riverine fluxes (at least 1.8 times larger than that of Redfield,

Ludwig et al., 2009; Koçak et al., 2010) it has been suggested that the Eastern Mediterranean receives excessive amounts of dissolved inorganic nitrogen and this unbalanced inputs may result in even more phosphorus deficiency (Ludwig et al., 2009; Koçak et al., 2010) whilst the atmospheric deposition of reactive nitrogen may cause accumulation of nitrogen in water column (Jickells et al., 2017). Very little research has focused on the importance of water- soluble organic nitrogen inputs to marine productivity in the Eastern Mediterranean (Mace et al., 2003a; Violaki and Mihalopoulos, 2010; Violaki et al., 2010). Hence, the unique contributions of the current study will be to (i) define the temporal variability of atmospheric water-soluble organic nitrogen, (ii) assign the origin of the water-soluble organic nitrogen, (iii) assess the influence of mineral dust on water-soluble organic nitrogen and (iv) enhance our knowledge of the quantitative dry and wet deposition for water-soluble organic nitrogen and its possible influence on marine productivity in the North Eastern Mediterranean.

These will be achieved by using the acquired data from the analyses for water soluble inorganic and organic nitrogen species of a series of size fractionated aerosol (coarse and fine)

and rain samples collected from March 2014 to April 2015 from the northern coast (Erdemli,

Turkey) of the Levantine Basin, Eastern Mediterranean.

**2. Material and Methods**

**2.1. Sampling Site Description**

Aerosol and rain sampling were carried out at a rural site located on the coast of the

Eastern Mediterranean, Erdemli, Turkey ($36°33′54″$ N and $34°15′18″$ E). The sampling tower (above sea level ~ 22 m, ~ 10 m away from the sea) is situated at the Institute of Marine

Sciences, Middle East Technical University (IMS-METU). Its immediate vicinity is surrounded by cultivated land to the north and to the south of the Northern Levantine Basin.

Although the site is not under the direct influence of any industrial activities (soda and fertilizer), the city of Mersin with a population of around 800.000 is located 45 km to the east of the sampling site (Kubilay and Saydam, 1995; Koçak et al., 2012) and hence aerosol and rainwater samples may have been influenced by aforementioned regional anthropogenic activities when air mass transported from the east.

**2.2. Sample Collection and Preparation**

*Aerosol:* A Gent type stacked filter unit (SFU) was used to collect aerosol samples in two size fraction (coarse: d = 10-2.5 μm and fine: d < 2.5 μm) (for more details see Hopke et al., 1997;

Koçak et al., 2007). Briefly, the first section of the filter holder was loaded with an 8 μm pore size polycarbonate filter (Whatman Track Etched 111114, circle diameter: 47 mm), whilst the second section was loaded with a 0.4 μm pore size polycarbonate filter (Whatman Track

Etched 111107, circle diameter: 47 mm). The cassette unit was then placed into the cylindrical cassette holder, which is designed to prevent the intrusion of particles larger than

10 μm when the sampler is operated at a flow rate of 16.0-16.5 L/min. Daily (24 hours)

temporal sample resolution was carried out. Operational blank filters were processed in the same way as the collected samples with the exception that no air was passed through the filters. In order to minimize any possible contamination, the filter loading and unloading were achieved in a laminar airflow cabinet.

The aerosol sampling campaign commenced in March 2014 and ended in April 2015.

During the sampling period, a total of 674 aerosol samples in two size fractions (coarse = 337; fine = 337) were obtained. The observational coverage of the aerosol sampling period was 80

%. The observational coverage for winter, spring, summer and fall was respectively 60 %, 92

%, 81 % and 79 %. The seasonal observational coverage, after applying a precision value of

0.3 (for more details see section 2.4 and Eq.4), was found to be comparable for winter (49 %), spring (53 %), summer (51 %) and fall (52 %).  The sampling was terminated from time to time due to technical malfunction of the SFU and/or cleaning procedure of the sampling apparatus.

*Rain:* Rainwater samples were collected using an automatic Wet/Dry sampler (Model ARS 1000, MTX Italy). A total of 23 rain samples were collected during the sampling period. After each rain event, the rainwater samples were immediately transferred to the laboratory for filtration (0.4 μm Whatman, polycarbonate filters). Operational blanks for rain samples were taken by using 100 mL of Milli-Q water after cleaning the HDPE buckets with phosphate free detergent, HCl (10 %) and Milli-Q water (3 times).

[revised manuscript text omitted]

WSON mean concentrations. Consequently, the presentation of the general characteristics of the data includes all negative concentrations (see Table 1) whilst the values presented in Table

1 will be used for calculating dry and wet deposition. It has been stated that the uncertainty in

WSON concentrations results from the additions of errors such as oxidation efficiency of method, sampling material, storage of samples and usage of preservative (Cape et al., 2011).

These authors have particularly pointed out that low precision for samples with low concentrations of WSON and high levels of WSIN (see Eq. 2). Although, the calculation of precision for WSON is very difficult owing to aforementioned errors, Hansell (1993) has proposed estimation of precision for WSON exclusively relaying on measured WSTN and

WSIN concentrations. Consequently, in order to evaluate the variability in the aerosol WSON

[revised manuscript text omitted]

adsorption, acid-base reaction) and/or between mineral dust and organic nitrogen compounds.

**3.5. Impact of Airflow on WSON**

Arithmetic mean concentrations together with corresponding standard deviations for water-soluble nitrogen species and nssCa$^{2+}$ in aerosol samples according to categorized air mass sectors (at 1 km) are presented in Table 5. WSON concentrations for Middle East, North

Africa and Turkey were comparable and arithmetic mean values were respectively 33, 36 and

32 nmol m$^{-3}$. Correspondingly, mean WSON concentrations for Eastern Europe, Western

Europe and Mediterranean Sea were 26, 26 and 22 nmol m$^{-3}$, being at least 1.2 times lower than those observed for Middle East, North Africa and Turkey (Mann-Whitney U test, $p <$

0.05). Coarse mode contributions of WSON for air flow from Middle East (61 %), North

Africa (58 %) and Turkey (63 %) ranged from 58 to 63 %. However, lower coarse mode contributions were observed when air flow originated from Eastern Europe (49 %), Western

Europe (48 %) and Mediterranean Sea (27 %). The highest $NO_3^-$ concentrations were associated with airflow from North Africa and Turkey with a value of 18 and 15 nmol N m$^{-3}$, respectively, and there was a statistically significant difference compared to the remaining air mass sectors ($p > 0.05$). The mean concentrations of $NO_3^-$ for air masses derived from North

Africa and Turkey was at least 1.3 times larger than those calculated for the Middle East,

Eastern Europe, Western Europe and Mediterranean Sea air sectors ($p > 0.05$). $NH_4^+$ had the highest concentration under the influence of airflow derived from Turkey. For this airflow, detected concentration was 1.5-2.4 times greater than those calculated for other air masses sectors. The Mann-Whitney test showed that there was a statistically significant difference in the $nssCa^{2+}$ concentrations. Arithmetic mean concentrations of $nssCa^{2+}$ in the Middle East and

North Africa were approximately 2 times higher compared to the remaining air masses. As expected, these two airflows were primarily influenced by crustal material due to sporadic dust events originating from deserts located in North Africa and the Middle East.

**3.6. Source Apportionment for WSON in Aerosol**

A number of studies have discussed the possible sources of WSON in aerosol material by applying either simple correlation analyses (Mace et al., 2003a; Violaki and Mihapoulos,

2010; Ho et al., 2015) or multivariate factor analyses (Chen and Chen, 2010), including PMF

(Chen et al., 2010). Usage of correlation analyses is useful when the number in samplepopulations are limited however; large datasets are required in order to carry out PMF and

[revised manuscript text omitted]

37.3 mmol N m$^{-2}$ yr$^{-1}$) was found to decrease about 45 % compared to the value reported by

Koçak et al. (2010, DIN = 70 mmol N m$^{-2}$ yr$^{-1}$). The reason of this decrease is out the scope of this article; nonetheless, there is a need to understand how DIN flux changed from the beginning of 2000s to 2015.

**4. Summary**

In the current study, water-soluble organic nitrogen in aerosol and rain samples obtained over the Eastern Mediterranean has been investigated. From this investigation the following summary may be made:

1) Of the nitrogen species, aerosol WSON (23.8 ± 16.3 nmol N m$^{-3}$) exhibited the highest arithmetic mean, followed by ammonium (23.3 ± 14.4 nmol N m$^{-3}$) and then nitrate (17.9 ± 15.7 nmol N m$^{-3}$). Aerosol WSON was mainly associated with coarse particles (66

%). The WSTN was equally influenced by WSON and NH$_4^+$, each contributing 37 and 35 %, respectively, whereas the contribution to WSTN of NO$_3^-$ was 28 %. In rainwater, the VWM

concentrations of water-soluble nitrogen species were comparable. WSON and NO$_3^-$

accounted for 29 and 32 % of the WSTN whilst NH$_4^+$ elucidated 39 % of the WSTN.

2) Aerosol WSON concentrations exhibited large variations from one day to another day. Generally, lower concentrations were observed during rainy days. Higher concentrations of aerosol WSON were associated with different airflow. The three highest concentrations were related to (i) mineral dust transport from Sahara and the Middle East deserts, (ii)

north/north westerly airflow from Turkey's largest cultivated plain, Konya and (iii) mid-range pollution transport from the Turkish coast.

3) Influence of mineral dust transport on aerosol WSON concentrations was assessed.

The crustally derived $nssCa^{2+}$ and anthropogenic $NO_3^-$ for dust events had arithmetic mean of

95.8 nmol $m^{-3}$ and 26.1 nmol N $m^{-3}$ which were almost four and two times higher than those of observed for non-dust events. The arithmetic mean of WSON (38.2 nmol $m^{-3}$) for dust events was 1.3 times higher compared to that observed for non-dust events (29.4 nmol $m^{-3}$).

4) Source apportionment suggested that aerosol WSON was mainly originated from anthropogenic sources including agricultural (43 %), secondary aerosols (20 %) and nitrate (22%), whereas, the two natural sources crustal material (10 %) and sea salts (5%) contributed

15 % to the WSON.

5) The total atmospheric deposition of water-soluble nitrogen (57.8 mmol N $m^{-2}$ $yr^{-1}$

$^1$) was mainly via wet deposition (36.7 mmol N $m^{-2}$ $yr^{-1}$). In contrast the atmospheric fluxes of

WSON and $NO_3^-$ were equally influenced by the dry and wet deposition modes. On average,

WSON accounted for 36 % of the total atmospheric deposition of WSTN. From the beginning of 2000s to 2015, the atmospheric deposition of the dissolved inorganic nitrogen declined about 45 %, as a consequence there is a need to understand how DIN flux changed.

**Appendix A**

In this section, the authors briefly summarize the main features of the positive matrix factorization (PMF).

PMF receptor model was described in detail by Paatero and Tapper (1994), Paatero (2007) and EPA PMF 5.0 User Guide. The details of the algorithm are also provided by Paatero (2007). This multivariate tool decomposes data matrix (X: n rows in other words number of samples and m columns: number of species) into two matrices: (i) source contributions $(G = n \, x \, p)$ and (ii) source profiles $(F = p \, x \, m)$. This can be given as follow

$$X = GF + E$$

where E and p denote the residual part and the number of factors extracted, respectively.

In order to run PMF, two input files are needed: (i) concentration and (ii) uncertainty. The first file includes concentrations whilst the second files contains uncertainty for each species. Uncertainty for PMF application can be calculated by different approaches such as ad hoc formula (Antilla et al., 1995), fixed fraction of the concentration (Paatero et al., 2014) or more complicated way as proposed by Polissar et al. (1998). No matter how it is calculated, if uncertainty is too high for one parameter, species will be categorized as bad by the PMF. For example, the precision of WSON for this study was found to be almost 3 times than that of arithmetic mean. If one uses the Eq.3 to calculate uncertainty of WSON for each data point, then it will be omitted by PMF owing to very high uncertainty values. Consequently, there will be no source apportionment for WSON. In order to obtain reasonable factor profiles for WSON, two step procedure was proposed. First, the usage of Eq.3 to eliminate WSON samples when their corresponding precisions are lower than mean R value of 0.3 (see Eq.4). Second, set the uncertainty to higher value for WSON (15 %) compared to the remaining species (5 %) since WSON inevitably exhibits very low precision (see Eq.1 and Eq.3).

After base run one has to estimate the quality of the obtained results from PMF (for more details see EPA PMF 5.0 user guide and Paatero et al., 2014). Base Model Displacement (DISP), Bootstrap (BS) and Bootstrap Displacement (BS-DISP) methods are the main tools to assess quality. It has been exhibited that three methods complement each other (for more details see Paatero et al., 2014). EFA PMF 5.0 provides aerosol data obtained from Baltimore and guides the applicant step by step to robustly use the source apportionment program of

EPA PM 5.0. More details are given by EFA 5.0 user guide and it is accessible to the scientific community.

The authors declare that they have no conflict of interest.

**Acknowledgments**
This work was mainly supported by The Scientific and Technological Research Council of
Turkey (TUBITAK). Required data were collected within the framework of the TUBITAK
113Y107 project. This study was also supported by the DEKOSIM (Center for Marine
Ecosystem and Climate Research) Project (BAP-08-11-DPT.2012K120880) funded by
Ministry of Development of Turkey. We would like to thank to Ersin Tursak, Pınar Kalegeri
and Merve Açıkyol for helping during sample collection and analysis. Aerosol optical
thickness, angstrom component and aerosol index values used in this study were produced
with the Giovanni online data system, developed and maintained by the NASA GES DISC.
We also acknowledge the MODIS and OMI mission scientists and associated NASA
personnel for the production of the data used in this research effort. The authors would like to
thank the two anonymous reviewers for their helpful comments which greatly improved the
submitted manuscript.

[revised manuscript text omitted]

Krom, M. D., Emeis, K-C., Van Cappellen, P.: Why is the Eastern Mediterranean phosphorus limited? *Prog. in Oceanogr., 85,* 236–244, 2010.

Kubilay, N. and Saydam, C.: Trace elements in atmospheric particulates over the Eastern Mediterranean: concentration, sources and temporal variability. *Atmos. Environ., 29,* 2289-2300, 1995.

Kubilay N., Nickovic S., Moulin C., Dulac F.: An illustration of the transport and deposition of mineral dust onto the eastern Mediterranean. *Atmos. Environ., 34,* 1293-1303, 2000.

Lee, E., Chan, C.K., Paatero, P.: Application of positive matrix factorization in source apportionment of particulate pollutants in Hong Kong. *Atmos. Environ., 33,* 3201–3212, 1999.

Lim, JM, Lee, JH, Moon, JH, Chung, YS, Kim, KH,. Source apportionment of $PM_{10}$ at a small industrial area using positive matrix factorization, *Atmos. Res.,* 95,88-100, 2010.

Ludwig, W., Dumont, E., Meybeck, M., and Heussner, S.: River discharges of water and nutrients to the Mediterranean and Black Sea: Major drivers for ecosystem changes during past and future decades? *Prog. in Oceanogr., 80 (3-4),* 199–217, 2009.

Mace, K.A., Kubilay, N. , Duce, R.A.: Organic nitrogen in rain and aerosol in the eastern Mediterranean atmosphere: An association with atmospheric dust. *J. Geophys. Res., 108, D10,* 4320, 2003a, doi:10.1029/2002JD002997.

Mace, KA, Duce RA, Tindale NW.: Organic nitrogen in rain and aerosol at Cape Grim, Tasmania Australia. *J. Geophys Res., 108*, 4338, 2003b.

Mace, KA, Artaxo, P., Duce, R.: Water-soluble organic nitrogen in Amazon Basin aerosols during the dry (biomass burning) and wet seasons. *J. Geophys. Res. 108*, 4512, 2003c, http://dx.doi.org/10.1029/2003JD003557.

Markaki, Z., Oikonomou, K., Koçak, M., Kouvarakis, G., Chaniotaki, A., Kubilay, N., Mihalopoulos, N.: Atmospheric deposition of inorganic phosphorus in the Levantine Basin, eastern Mediterranean: Spatial and temporal variability and its role in seawater productivity. *Limnol. Oceanorg.,, 48 (4),* 1557–1568, 2003. doi: 10.4319/lo.2003.48.4.1557

Markaki, Z., Loÿe-Pilot, M. D., Violaki, K., Benyahya, L., Mihalopoulos, N.; Variability of atmospheric deposition of dissolved nitrogen and phosphorus in the Mediterranean and possible link to the anomalous seawater N/P ratio. *Mar. Chem., 120(1–4),* 187–194, 2010. doi: 10.1016/ j.marchem.2008.10.005.

Mihalopoulos, N., Kerminen, V., M., Kanakidou, M., Berresheim, H., Sciare, J.: Formation of particulate sulfur species (sulfate and methanesulfonate) during summer over the Eastern Mediterranean: a modelling approach. *Atmos. Environ. 41,* 6860-6871, 2007.

Miller, J.: The nitrogen content of rain falling at Rothamsted. *J. Agricul. Sci., 1,* 280-303, 1905.

Miyazaki, Y., Kawamura, K., Jung, J., Furutani, H., Uematsu, M.: Latitudinal distributions of organic nitrogen and organic carbon in marine aerosols over the western North Pacific, *Atmos. Chem. Phys., 11,* 3037–3049, 2011. doi:10.5194/acp-11-3037-2011.

Neff, J. C., Holland, E. A., Dentener, F. J., McDowell, W. H., and Russell, K. M.: The origin, composition and rates of organic nitrogen deposition: A missing piece of the nitrogen cycle. *Biogeochemistry, 57*, 99–136, 2002.

Paatero, P., Tapper, U.: Positive matrix factorization: a non-negative factor model with optimal utilization of error estimates of data value. *Environmetrics, 5,* 111–126, 1994.

Peierls, B. L. and Paerl, H. W.: Bioavailability of atmospheric organic nitrogen deposition to coastal phytoplankton. *Limnol. Oceanogr., 42,* 1819–1823, 1997.

Paatero, P.: End User's Guide to Multilinear Engine Applications, 2007. ftp://ftp.clarkson.edu/pub/hopkepk/pmf/.

Paatero, P., Eberly, S., Brown, S. G., Norris, G. A.: Methods for estimating uncertainty in factor analytic solutions. *Atmos. Meas. Tech., 7,* 781–797, 2014. doi:10.5194/amt-7-781-2014.

Powley, H. R., Krom, M. D., Van Cappellen, P.: Understanding the unique biogeochemistry of the Mediterranean Sea: Insights from a coupled phosphorus and nitrogen model. *Global Biogeochem. Cy., 31,* 1010-1031, 2017. doi: 10.1002/2017GB005648.

[revised manuscript text omitted]

---

## Author Response (AR1)

Dear Dr. Evangelos Gerasopoulos,

The authors would like to thank the two reviewers for their very helpful and informative comments. Please find below our responses to all the comments made by the two reviewers. We are confident that the modified manuscript as a result of our response to the reviewer's comments will be a clearer and more focused document.

Please find below our answers and response to comments.

Yours Faithfully,

Dr. Mustafa Koçak

**Reviewer 1**

Q1: There have been some recent reviews of WSON which the authors might reference (e.g. Cape et al., 2011 Atmos. Res 102,30-48) since they summaries much of the material and offer a somewhat wider perspectives and more recent information on the composition of WSON.
*A1: Cape et al., 2011 was also used for defining the composition of WSON, including vehicle exhaust, cooking, algal blooms and degraded proteins (see lines 48,49, 50 and 52).*

Q2: There is also now a global model of WSON (Kanakidou et al., 2012 Global Biogeochem. Cycl.26, doi10.1029/2011GB004277) which has contributed to an updated global nitrogen cycle revising the Duce et al 2008 paper cited (Jickells et al., 2017 Global Biogeochem. Cycl.31, doi10.1029/2016GB005586).
*A2: Jickells et al. 2017 was used to update change in reactive nitrogen and reactive anthropogenic organic nitrogen from mid 1800 to 2000s (see lines 68-70).*

Q3: Line 60, while amines will neutralise acids, it is not obvious the rest of WSON will.
*A: The sentence was changed as follow 'Similar to ammonium, 'some' organic nitrogen species such as urea and amines have acid-neutralizing capacities (Ge et al., 2011)'. Urea was kept in the text since it shows slightly alkaline character. Furthermore, amino-acids such as Lysine, Histidine and Arginine exhibit alkaline character (see line 58).*

Q4: Line 62 I don't think Twohy discusses WSON.
*A4: Twohy et al. (2005) has mentioned about organo-nitrogen compounds. To quote Twohy et al. 'Our analysis indicates that organic species do not have to be mixed with inorganic particles to act as CCN, and that organo-nitrogen compounds nucleate cloud droplets in the Indian Ocean. If these particles are present in other polluted areas, they could contribute substantially to the global indirect aerosol effect' (conclusion page 3, 12th paragraph). Therefore, if these organo-nitrogen compounds nucleate cloud droplets, they would be water-soluble.*

Q5: Line 80 and later on, there is really pretty clear Evidence that the Eastern Mediterranean is P limited. There is a vast body of work by Krom and colleagues that supports this (see most recently Pawley et al 2017Global Biogeochem.Cycl.31, 1010-1031 and the earlier summary in Krom et al 2010Prog. In Oceanography 85,236-244) and my reading of the Yücel 2017 paper does not actually contradict this view.

*A5: This issue was clarified as suggested (see lines from 76 to 87).*

Q6: Line81-2 It is mentioned a little bit later on, but not here, that Mace et al have reported WSON from exactly the same site as the study here. This should be noted here and also in section 3.1.

*A6: Reference was noted as advised (see lines 88-89 and 296-297).*

Q7: Note also the reference list lacks dates and while in the text the authors refer to Mace et al a,b and c, these are not identified in the references by these letters.

*A7: Correction was made considering Mace et al 2003a, b and c (see lines from 783 to 790).*

Q8: Analytical Methods. In general the results seem to be of good quality, although there is no mention of how blanks were determined (i.e. what procedures were used to create blank samples for analysis), what standards were used in analysis and whether any certified reference materials were used.

*A8: As stated before Operational blank filters for aerosol were processed in the same way as the collected samples with the exception that no air was passed through the filters. Information about blanks for rain was added to text: Operational blanks for rain samples were taken by using 100 mL of Milli-Q water after cleaning the HDPE buckets with phosphate free detergent, HCl (10 %) and Milli-Q water (see lines from 139 to 141).Details about standard, recovery analysis and quality assurance for WSTN was given in Materials and Methods section (see lines from 169 to 178).*

Q9: I do not really understand what the sentence line163-4 about blanks being <10 % means, Is this true for all ions?

*A9: Sentence was clarified by using 'for all' (see line 189-190)*

Q10: On line163 20 ppb is ambiguous, is it as ppb nitrogen and Why not use molar units as elsewhere in the paper?

*A10: Molar unit was used as suggested (see line 179).*

Q11: Section2.4 discusses the quite well known challenges of estimating WSON and its relatively low precision as a derived quantity (see Cape et al for instance). The precision of WSON depends a lot on the Relative concentrations of the three components of the total nitrogen analysis, so it is not possible really to quote a single number. The authors discussion e.g. lines170-174 and 175 (and lines 221-222) does not really explain what they actually estimate the precision to be.

*A11: Modification was made as suggested (see lines from 208 to 216).*

Q12: The use of PMF (which I am no expert on) here seems to require provision of precision estimates, but I do not understand how the arbitrary thresholds used here (line185-7) were arrived at or how sensitive there results are to these values.

*A12: Appendix was added in order to clarify uncertainty estimates (see lines from 602 to 633). As it is well known, it impossible to include negative values when one carries out multivariate statistics. Application of PMF without negative values, yielded worse slope than that of obtained by using threshold. For instance, the slope of the estimated WSON against measured WSON was 30 % less than unity. Furthermore, DISP error estimates showed that there were factor swaps and significant change in Q during DISP. In other words, PMF application without threshold exhibited that solutions were not robust.*

Q13: Section2.6 As I understand it PMF is a form of principal component analysis and hence is an appropriate tool for this kind of source apportionment. I would suggest the authors may be better putting an explanation of the principal of the method here and putting the highly technical discussion in to some sort of appendix, because I think many readers will not really be able to follow this section.

*A13: We agree with the comment. Consequently, appendix was added into the text (see lines from 602 to 6633).*

Q14: Section 3.1 and 3.2 I Wonder if these sections could be shortened a bit given that the results are broadly in line with other work in this region.

A14: *As can be seen from the Table 2, there are only two references in the region (Mace et al., 2003 and Violaki and Mihalopoulos, 2010). If one relies on these two references, the scientific discussion and comparison would be insufficient to enrich the findings/arguments about WSON. Thus, the 3.1 and 3.2 were preserved as is.*

Q15: Line 304-307 I do not disagree with the interpretation here, but it is worth noting that this does carry the implicit assumption that land based sources dominate the emission of WSON.

A15: *We agree with the comment. '(iii) small contributions from non-land based local emissions such as sea salt and algal blooms' was added as third reason.*

Q16: Section 3.3. this section is very general and the issue is approached in a more quantitative manner in 3.5 and 3.6, so I wonder if the section could be shortened.

A17: *We agree that more detail was given in 3.5 and 3.6, however, temporal variability (Section 3.3) includes both daily and seasonal variability and it is only one and a half page. It was kept as is.*

Q17: Section 3.4 Mace et al suggested that the Saharan Dust was a major source of WSON at this site and they did this I think by a correlation between $nssCa^{2+}$ and WSON. Here the association with dust seems to be weaker but the discussion does not really address this point, but simply notes there is an association with dust. This could be discussed further.

A17: *Regarding Mace et al., 2003a this summary may be made. The obtained samples only covers dust period. Indeed, there was a strong correlation between $nssCa^{2+}$ and WSON ($R^2$ =075). On the other hand, there was also a strong correlation between nitrate and WSON (R2 = 0.69) (Mace et al., 2003a, page 5-7, Table 3). Subsequently, even during the dust period (from March 22 to May 4) it seems that WSON was almost equally impacted by mineral dust (not sure only from Sahara, see below) and man-made nitrate. Moreover, I am not sure whether all dust events were originated from Sahara Desert. For example, Figure 2a (Mace et al., 2003a, page 5-3) shows that on 5 April 2000 the Erdemli site was under the influence of the Middle East (air masses arriving at 1000 and 900 hPa) and Sahara (air masses arriving at 700 and 500 hPa) Deserts at the same time. However, such an interpretation in the text would be impolite since the authors only considered Saharan Dust in spite of strong correlation between nitrate and WSON and air masses back trajectories.*

Q18: Section 3.5 In Table 5 the WSON and other parameters are classified into 5 groups, but in the text here the discussion splits the data in to two. It would be easier for the reader if the manuscript discussion and the tables did one or other of these, rather than mix them up in this way.

*A18: Modification in this section was made as advised (see lines from 472 to 478).*

Q19: Section 3.6 As noted earlier I am no expert PMF. The striking thing for me from Figure6 and the discussion, is that WSON does not resolve in any simple way in to any of the components identified, emphasizing the multiplicity of sources that it has, and this is particularly striking within such a large data set. I would also query the interpretation of what the associations mean (lines 469-474). The authors interpret the results in terms of formation mechanisms, but an alternative explanation might be emission sources.

*A19: We agree with the suggestion. Therefore, Factors 1, 2 and 5 were respectively attributed to ammonium-bisulfate (regional emissions), nitrate (combustion) and soil re-suspension (see lines 515-518 and 522-524).*

Q20: Section 3.7As noted earlier the Eastern Mediterranean appears to be phosphorus limited. If this is the case then the addition of nitrogen will not necessarily stimulate any

Additional primary production, but rather contribute to the high N/P ratio (see earlier) Krom and Pawley references) and so the hypothesis behind the calculation (line500-

509) is flawed and the conclusions about the impacts on new production are incorrect.

Section4. This is really a summary and not a conclusion and simply repeats the earlier material.

*A20: Please see A5. The last paragraph of section 3.7 was removed from the text as suggested by Reviewer 2. We agree that the section 3.4 is not a conclusion. Thus, the summary was used instead of conclusion (see line 560).*

**Reviewer 2**

Q1: Aerosol WSON concentrations exhibited large temporal variations mainly due to rain and the origin of air mass flow. Rain scavenges all compounds why preferably WSON?

*A1: I agree. Rain scavenges all compounds. Meteorology was used instead of rain (see page 1, line 21).*

Q2: $NO_3$ is a secondary aerosol?

*A2: $NO_3$ is a secondary aerosol (please see A10).*

Q3: Considering the Cilician Basin, the atmospheric water soluble nitrogen flux would sustain 33% and 76% of the new production in the associated coastal and open waters, respectively. I suggest to remove this part or to be more cautious.

*A3: The sentence (the last paragraph of the section 3.7) was removed as suggested.*

Q4: Did the samples were uniformly distributed all over seasons?

*A4: Information about the seasonal observational coverage was added to the experimental section (see answer to question 8).*

Q5: Did the authors perform recovery experiments with well-known mixtures of organic and inorganic compounds?

*A5: Recovery experiments were carried out by using nitrate, ammonium, urea and mixture of all three species. Furthermore, the performance of our laboratory was studied by participating Quality Assurance of Information for Marine Environmental Monitoring in Europe (QUASIMEME) program. More information about recovery experiments was supplemented (see lines from 169 to 179).*

Q6: Please also report median in addition to average. Also how mean and median values changes if all data are considered (sensitivity test).

*A6: Median values were typed in Table 2. Number of samples were also included. Table 2 shows statistical summary for all samples (see line 287 and 288). Median values for WSON, $NO_3^-$, $NH_4^+$ and WSTN were respectively 10 %, 20 %, 40 % and 10 % lower than those that of arithmetic means.*

Q7: What is the seasonality of coarse/fine ratio of WSON? Also how this ratio varies as a function of air masses origin or better in dust vs non-dust samples?

*A7: Details about coarse/fine ratio of WSON for each season, air mass and dust/non-dust were given in Table 4 and 5 (also see lines 408-410, 458-460 and 477-480).*

Q8: At Table 4, the number of samples reported is only 216, whereas at line 120 they report analysis of 337 results. Why this difference? They correspond to samples with precision larger than 0,3? If yes what is the seasonal representativity of these 216 samples?

*A8: A total of 216 aerosols corresponded to samples with precision larger than 0.3. The observational coverage for winter, spring, summer and fall was respectively 60 %, 92 %, 81 % and 79 %. The seasonal observational coverage, after applying precision value of 0.3 (for more details see section 2.4 and Eq.4), was found to be comparable for winter (49 %), spring (53 %), summer (51 %) and fall (52 %) (see lines 129-132).*

Q9: In rain water how many values have been omitted with precision lower than 0,3? What is the temporal variability of the omitted data?

*A9: During the calculations of the dry and wet deposition, the values presented in Table 1 were utilized. Therefore, none of the aerosol and rain samples was omitted (see lines 208-209 and 221-222).*

Q10: Sources attribution (PMF). Using only IC data, source attribution of WSON is highly subjective. No ancillary data such as metals or OC/EC, Nox are available? Why no $NH_4$ is founding agricultural factor? Given the compounds associated with this factor better assign it to soil re-suspension. Also factor A should better attributed to long-range transport (regional sources) due to the presence of $NH_4$ and $SO_4$. Similarly factor B with the presence mainly of $NO_3$ could be better attributed to anthropogenic sources (combustion) Line 487, better replace water-soluble nitrogen by WSTN.

*A10: I agree that usage of ancillary data such metals and OC/EC during PMF would yield better results. Unfortunately there was no ancillary data such as metals, OC/EC and NOx. However, factor 1, 2, 3 and 4 were comparable to those obtained for Erdemli by using water-soluble ions and metals (Koçak et al., 2009). Factors 1, 2 and 5 were respectively attributed to ammonium-bisulfate (regional), nitrate (combustion) and soil re-suspension as suggested. Small amount of $NH_4$ (3.7 %) was associated with factor 5. As stated in section 3.1, 96 % of the $NH_4$ was originated from fine mode. Moreover, considering the first 20 % of the highest loadings in factor 5 (re-suspension), there was strong relationship between WSON and $NH_4$ (r = 0.64). Thus, it sees that $NH_4$ was reasonably associated with re-suspension. . Water-soluble nitrogen was replaced by WSTN (see 515-518, 522-524 and Figure 6).*

[revised manuscript text omitted]

The Mediterranean Sea is characterized by oligotrophic surface waters with Low Nutrient Low Chlorophyll (LNLC) regions. This has been attributed to mainly anti-estuarine (reverse thermohaline) circulation (Hamad et al., 2005). The Eastern Mediterranean (25) has higher molar N/P ratios than those observed in the Western Mediterranean (22) and the Redfield ratio (Krom et al., 2004; Yılmaz and Tuğrul, et al., 1998). Generally, the primary productivity in the Eastern Mediterranean is phosphorous limited (Krom et al., 1991; Krom et al., 2010; Powley et al., 2017). Depending on season, the limitation by nitrogen or co-limitation by nitrogen and phosphorus in the Eastern Mediterranean have been reported (Yücel, 2013; Yücel, 2017 and references therein). Based on molar N/P ratios in the atmospheric input (order of magnitude higher than that of Redfield, Markaki et al., 2003, 2010 ; Koçak et al., 2010) and riverine fluxes (at least 1.8 times larger than that of Redfield,

Ludwig et al., 2009; Koçak et al., 2010) it has been suggested that the Eastern Mediterranean receives excessive amounts of dissolved inorganic nitrogen and this unbalanced inputs may result in even more phosphorus deficiency (Ludwig et al., 2009; Koçak et al., 2010) whilst the atmospheric deposition of reactive nitrogen may cause accumulation of nitrogen in water column (Jickells et al., 2017). Very little research has focused on the importance of water- soluble organic nitrogen inputs to marine productivity in the Eastern Mediterranean (Mace et al., 2003a; Violaki and Mihalopoulos, 2010; Violaki et al., 2010). Hence, the unique contributions of the current study will be to (i) define the temporal variability of atmospheric water-soluble organic nitrogen, (ii) assign the origin of the water-soluble organic nitrogen, (iii) assess the influence of mineral dust on water-soluble organic nitrogen and (iv) enhance our knowledge of the quantitative dry and wet deposition for water-soluble organic nitrogen and its possible influence on marine productivity in the North Eastern Mediterranean.

These will be achieved by using the acquired data from the analyses for water soluble inorganic and organic nitrogen species of a series of size fractionated aerosol (coarse and fine)

and rain samples collected from March 2014 to April 2015 from the northern coast (Erdemli,

Turkey) of the Levantine Basin, Eastern Mediterranean.

**2. Material and Methods**

**2.1. Sampling Site Description**

Aerosol and rain sampling were carried out at a rural site located on the coast of the

Eastern Mediterranean, Erdemli, Turkey ($36°\,33'\,54''$ N and $34°\,15'\,18''$ E). The sampling tower (above sea level $\sim 22$ m, $\sim 10$ m away from the sea) is situated at the Institute of Marine

Sciences, Middle East Technical University (IMS-METU). Its immediate vicinity is surrounded by cultivated land to the north and to the south of the Northern Levantine Basin.

Although the site is not under the direct influence of any industrial activities (soda and fertilizer), the city of Mersin with a population of around 800.000 is located 45 km to the east of the sampling site (Kubilay and Saydam, 1995; Koçak et al., 2012) and hence aerosol and rainwater samples may have been influenced by aforementioned regional anthropogenic activities when air mass transported from the east.

**2.2. Sample Collection and Preparation**

*Aerosol:* A Gent type stacked filter unit (SFU) was used to collect aerosol samples in two size fraction (coarse: d = 10-2.5 μm and fine: d < 2.5 μm) (for more details see Hopke et al., 1997;

Koçak et al., 2007). Briefly, the first section of the filter holder was loaded with an 8 μm pore size polycarbonate filter (Whatman Track Etched 111114, circle diameter: 47 mm), whilst the second section was loaded with a 0.4 μm pore size polycarbonate filter (Whatman Track

Etched 111107, circle diameter: 47 mm). The cassette unit was then placed into the cylindrical cassette holder, which is designed to prevent the intrusion of particles larger than

10 μm when the sampler is operated at a flow rate of 16.0-16.5 L/min. Daily (24 hours)

temporal sample resolution was carried out. Operational blank filters were processed in the same way as the collected samples with the exception that no air was passed through the filters. In order to minimize any possible contamination, the filter loading and unloading were achieved in a laminar airflow cabinet.

 The aerosol sampling campaign commenced in March 2014 and ended in April 2015.

During the sampling period, a total of 674 aerosol samples in two size fractions (coarse = 337; fine = 337) were obtained. The observational coverage of the aerosol sampling period was 80

%. The observational coverage for winter, spring, summer and fall was respectively 60 %, 92

%, 81 % and 79 %. The seasonal observational coverage, after applying a precision value of

0.3 (for more details see section 2.4 and Eq.4), was found to be comparable for winter (49 %), spring (53 %), summer (51 %) and fall (52 %).  The sampling was terminated from time to time due to technical malfunction of the SFU and/or cleaning procedure of the sampling apparatus.

*Rain:* Rainwater samples were collected using an automatic Wet/Dry sampler (Model ARS 1000, MTX Italy). A total of 23 rain samples were collected during the sampling period. After each rain event, the rainwater samples were immediately transferred to the laboratory for filtration (0.4 μm Whatman, polycarbonate filters). Operational blanks for rain samples were taken by using 100 mL of Milli-Q water after cleaning the HDPE buckets with phosphate free detergent, HCl (10 %) and Milli-Q water (3 times).

[revised manuscript text omitted]

Median values for WSON, $NO_3^-$, $NH_4^+$ and WSTN were respectively 10 %, 20 %, 40 % and

10 % lower than those of arithmetic means.  Among the nitrogen species, WSON exhibited the highest arithmetic mean, followed by ammonium and nitrate concentrations respectively.

The maximum concentration of WSON was estimated to be 79 nmol N m$^{-3}$ with a mean value and standard deviation of 23.8 ± 16.3 nmol N m$^{-3}$. The observed arithmetic was comparable to those reported by Mace et al. (2003a) for the same site. Approximately 66 % of the WSON

was associated with coarse particles, the remaining fraction (34 %) was present within the fine mode. A number of studies have reported the relative size distribution of WSON for the

Eastern Mediterranean marine aerosol  (Finokalia, Violaki and Mihalopoulos, 2010) and those observed at remote marine sites (Hawaii, Cornell et al., 2001; Tasmania, Mace et al., 2003b).

The aerosol WSON at Finokalia (68 %) and Hawaii were primarily found in the fine mode whilst WSON in the south Pacific marine aerosol (Tasmania) was mainly associated with the coarse fraction. It is likely that the WSON at Erdemli (a) is relatively less impacted by anthropogenic sources and/or (b) is more influenced by mineral dust transport and re- suspension of cultivated soil compared to that observed at Finokalia.

$NO_3^-$ and $NH_4^+$ aerosol concentrations ranged between 0.2-88.4 and 0.5-164.4 nmol N

$m^{-3}$ with mean values (standard deviations) of 17.9 (±15.7) and 23.3 (±24.4) nmol N $m^{-3}$. As expected, $NO_3^-$ was mainly associated with coarse particles, accounting for 87 % of the observed mean value while $NH_4^+$ was dominant in the by fine mode, contributing 96 % to the detected mean concentration. Similar results have been reported for Eastern Mediterranean marine aerosol (Bardouki et al, 2003; Koçak et al., 2007). The predominance of $NO_3^-$ in the coarse mode might be due to gaseous nitric acid or other nitrogen oxides reacting with alkaline sea salts and mineral dust particles. In contrast, the occurrence of $NH_4^+$ in the fine fraction is mainly as a result of the reaction between gaseous alkaline ammonia and acidic sulfuric acid (Mihalopoulos et al., 2007).

WSTN concentrations in aerosols varied between 9.7 and 176.5 nmol N $m^{-3}$ with an arithmetic mean value of 63.5± 32.0 nmol N $m^{-3}$, respectively. The mean WSTN

concentration being almost equally influenced by coarse (51 %) and fine particles (49 %).

Table 2 demonstrates the relative contributions of WSON, $NO_3^-$ and $NH_4^+$ to the WSTN in

$PM_{10}$. As can be deduced from the table, the WSTN concentration was equally influenced by

WSON and $NH_4^+$, each species contributing 37% and 35 %, respectively. In contrast the contribution of $NO_3^-$ to WSTN was found to be 28 %.

*Rain:* Volume-weighted-mean (VWM) concentrations of WSON, $NO_3^-$, $NH_4^+$ and WSTN in rainwater are presented in Table 2, along with the minimum and maximum concentrations as well as the relative contributions of WSON, $NO_3^-$ and $NH_4^+$ to WSTN. As can be deduced from table, VWM concentrations of each species were comparable, $NH_4^+$ exhibited the highest concentration with a value of 28.7 µmol N $L^{-1}$. The VWM concentration of WSON and $NO_3^-$

were 21.5 and 23.3 μmol N L$^{-1}$, respectively. Considering their relative contributions to

WSTN, WSON and NO$_3^-$ account 29 and 32 % of the WSTN whilst NH$_4^+$ represented 39 %

of the observed WSTN concentration in rainwater.

**3.2. Comparison of WSON in Aerosol and Rain with data from the Literature**

The concentrations of WSON in marine aerosols and rain samples collected from different sites located around the Mediterranean, Atlantic and Pacific regions are illustrated in

Table 3. Comparing the current WSON values with those reported in the literature is challenging due to (i) different applied sampling periods, sampling and measurement techniques and (ii) the high uncertainty associated with the estimation of WSON Furthermore, within the literature there is a lack of information defining the uncertainty of WSON though there is a substantial statistical knowledge. Keene at al. (2002) in particular, have highlighted the tendency in the literature to neglect negative values or substitute such values with zero instead when calculating the WSON from the difference between WSTN and WSIN. As these authors have highlighted the omission or substitution of such values inevitably would result in a positive bias in the WSON concentrations.

In general, the lowest concentrations in aerosols were found in those derived from remote or pristine marine environments. The WSON concentrations in the atmosphere over the Indian (Amsterdam Island: 1.0 nmol N m$^{-3}$, Violaki et al., 2015), Atlantic (Barbados: 1.3

nmol N m$^{-3}$, Zamora et al., 2011) and Pacific Ocean (Hawaii, Oahu: 4.1 nmol N m$^{-3}$, Cornell et al., 2001, Tasmania: 5.3 nmol N m$^{-3}$, Mace et al., 2003b) were at least 4 times less than those observed for Eastern Mediterranean (Erdemli: 23.8 nmol N m$^{-3}$, this study; Finokalia:

17.1 nmol N m$^{-3}$, Violaki and Mihalopoulos, 2010). These lower values might be attributed to (i) the absence of the strong anthropogenic sources in the vicinity of the sampling sites, (ii)

the dilution of the WSON originating from long range transport via both dry and wet deposition and/or (iii) small contributions from non-land based local emissions such as sea salt and algal blooms. 
[revised manuscript text omitted]
$^{-3}$). Percent contributions of coarse WSON for dust and non-dust events were almost identical, being 58 %

and 60 %, respectively.  A similar enrichment of WSON during dust events has been reported for Erdemli (Mace et al., 2003a; Yellow Sea (Shi et al., 210) and Finokalia (Violaki and

Mihalopoulos, 2010). In addition, Griffin et al. (2007) have demonstrated a significant difference between dust and non-dust events for bacterial and fungal colony forming units at

Erdemli, the former being much greater. Thus, it might be speculated that this enhancement during dust events can be due to (a) mineral dust borne microorganisms, (b) interaction (e.g.

adsorption, acid-base reaction) and/or between mineral dust and organic nitrogen compounds.

**3.5. Impact of Airflow on WSON**

Arithmetic mean concentrations together with corresponding standard deviations for water-soluble nitrogen species and nssCa$^{2+}$ in aerosol samples according to categorized air mass sectors (at 1 km) are presented in Table 5. WSON concentrations for Middle East, North

Africa and Turkey were comparable and arithmetic mean values were respectively 33, 36 and

32 nmol m$^{-3}$. Correspondingly, mean WSON concentrations for Eastern Europe, Western

Europe and Mediterranean Sea were 26, 26 and 22 nmol m$^{-3}$, being at least 1.2 times lower than those observed for Middle East, North Africa and Turkey (Mann-Whitney U test, p <

0.05). Coarse mode contributions of WSON for air flow from Middle East (61 %), North

Africa (58 %) and Turkey (63 %) ranged from 58 to 63 %. However, lower coarse mode contributions were observed when air flow originated from Eastern Europe (49 %), Western

Europe (48 %) and Mediterranean Sea (27 %). The highest $NO_3^-$ concentrations were associated with airflow from North Africa and Turkey with a value of 18 and 15 nmol N m$^{-3}$, respectively, and there was a statistically significant difference compared to the remaining air mass sectors ($p > 0.05$). The mean concentrations of $NO_3^-$ for air masses derived from North

Africa and Turkey was at least 1.3 times larger than those calculated for the Middle East,

Eastern Europe, Western Europe and Mediterranean Sea air sectors ($p > 0.05$). $NH_4^+$ had the highest concentration under the influence of airflow derived from Turkey. For this airflow, detected concentration was 1.5-2.4 times greater than those calculated for other air masses sectors. The Mann-Whitney test showed that there was a statistically significant difference in the $nssCa^{2+}$ concentrations. Arithmetic mean concentrations of $nssCa^{2+}$ in the Middle East and

North Africa were approximately 2 times higher compared to the remaining air masses. As expected, these two airflows were primarily influenced by crustal material due to sporadic dust events originating from deserts located in North Africa and the Middle East.

**3.6. Source Apportionment for WSON in Aerosol**

A number of studies have discussed the possible sources of WSON in aerosol material by applying either simple correlation analyses (Mace et al., 2003a; Violaki and Mihapoulos,

2010; Ho et al., 2015) or multivariate factor analyses (Chen and Chen, 2010), including PMF

(Chen et al., 2010). Usage of correlation analyses is useful when the number in sample- populations are limited however; large datasets are required in order to carry out PMF and

[revised manuscript text omitted]

WSON accounted for 36 % of the total atmospheric deposition of WSTN. From the beginning of 2000s to 2015, the atmospheric deposition of the dissolved inorganic nitrogen declined about 45 %, as a consequence there is a need to understand how DIN flux changed.

**Appendix A**

In this section, the authors briefly summarize the main features of the positive matrix factorization (PMF).

PMF receptor model was described in detail by Paatero and Tapper (1994), Paatero (2007) and EPA PMF 5.0 User Guide. The details of the algorithm are also provided by Paatero (2007). This multivariate tool decomposes data matrix (X: n rows in other words number of samples and m columns: number of species) into two matrices: (i) source contributions $(G = n \times p)$ and (ii) source profiles $(F = p \times m)$. This can be given as follow

$$X = GF + E$$

where E and p denote the residual part and the number of factors extracted, respectively.

In order to run PMF, two input files are needed: (i) concentration and (ii) uncertainty. The first file includes concentrations whilst the second files contains uncertainty for each species. Uncertainty for PMF application can be calculated by different approaches such as ad hoc formula (Antilla et al., 1995), fixed fraction of the concentration (Paatero et al., 2014) or more complicated way as proposed by Polissar et al. (1998). No matter how it is calculated, if uncertainty is too high for one parameter, species will be categorized as bad by the PMF. For example, the precision of WSON for this study was found to be almost 3 times than that of arithmetic mean. If one uses the Eq.3 to calculate uncertainty of WSON for each data point, then it will be omitted by PMF owing to very high uncertainty values. Consequently, there will be no source apportionment for WSON. In order to obtain reasonable factor profiles for WSON, two step procedure was proposed. First, the usage of Eq.3 to eliminate WSON samples when their corresponding precisions are lower than mean R value of 0.3 (see Eq.4). Second, set the uncertainty to higher value for WSON (15 %) compared to the remaining species (5 %) since WSON inevitably exhibits very low precision (see Eq.1 and Eq.3).

After base run one has to estimate the quality of the obtained results from PMF (for more details see EPA PMF 5.0 user guide and Paatero et al., 2014). Base Model Displacement (DISP), Bootstrap (BS) and Bootstrap Displacement (BS-DISP) methods are the main tools to assess quality. It has been exhibited that three methods complement each other (for more details see Paatero et al., 2014). EFA PMF 5.0 provides aerosol data obtained from Baltimore and guides the applicant step by step to robustly use the source apportionment program of

EPA PM 5.0. More details are given by EFA 5.0 user guide and it is accessible to the scientific community.

The authors declare that they have no conflict of interest.

**Acknowledgments**
This work was mainly supported by The Scientific and Technological Research Council of
Turkey (TUBITAK). Required data were collected within the framework of the TUBITAK
113Y107 project. This study was also supported by the DEKOSIM (Center for Marine
Ecosystem and Climate Research) Project (BAP-08-11-DPT.2012K120880) funded by
Ministry of Development of Turkey. We would like to thank to Ersin Tursak, Pınar Kalegeri
and Merve Açıkyol for helping during sample collection and analysis. Aerosol optical
thickness, angstrom component and aerosol index values used in this study were produced
with the Giovanni online data system, developed and maintained by the NASA GES DISC.
We also acknowledge the MODIS and OMI mission scientists and associated NASA
personnel for the production of the data used in this research effort. The authors would like to
thank the two anonymous reviewers for their helpful comments which greatly improved the
submitted manuscript.

[revised manuscript text omitted]

Krom, M. D., Emeis, K-C., Van Cappellen, P.: Why is the Eastern Mediterranean
phosphorus limited? *Prog. in Oceanogr., 85,* 236–244, 2010.

Kubilay, N. and Saydam, C.: Trace elements in atmospheric particulates over the
Eastern Mediterranean: concentration, sources and temporal variability. *Atmos. Environ., 29,*
2289-2300, 1995.

Kubilay N., Nickovic S., Moulin C., Dulac F.: An illustration of the transport and
deposition of mineral dust onto the eastern Mediterranean. *Atmos. Environ., 34,* 1293-1303,
2000.

Lee, E., Chan, C.K., Paatero, P.: Application of positive matrix factorization in source
apportionment of particulate pollutants in Hong Kong. *Atmos. Environ., 33,* 3201–3212, 1999.

Lim, JM, Lee, JH, Moon, JH, Chung, YS, Kim, KH,. Source apportionment of PM$_{10}$ at a small industrial area using positive matrix factorization, *Atmos. Res.,* 95,88-100, 2010.

Ludwig, W., Dumont, E., Meybeck, M., and Heussner, S.: River discharges of water and nutrients to the Mediterranean and Black Sea: Major drivers for ecosystem changes during past and future decades? *Prog. in Oceanogr., 80 (3-4),* 199–217, 2009.

Mace, K.A., Kubilay, N. , Duce, R.A.: Organic nitrogen in rain and aerosol in the eastern Mediterranean atmosphere: An association with atmospheric dust. *J. Geophys. Res., 108, D10,* 4320, 2003a, doi:10.1029/2002JD002997.

Mace, KA, Duce RA, Tindale NW.: Organic nitrogen in rain and aerosol at Cape Grim, Tasmania Australia. *J. Geophys Res., 108*, 4338, 2003b.

Mace, KA, Artaxo, P., Duce, R.: Water-soluble organic nitrogen in Amazon Basin aerosols during the dry (biomass burning) and wet seasons. *J. Geophys. Res. 108*, 4512, 2003c, http://dx.doi.org/10.1029/2003JD003557.

Markaki, Z., Oikonomou, K., Koçak, M., Kouvarakis, G., Chaniotaki, A., Kubilay, N., Mihalopoulos, N.: Atmospheric deposition of inorganic phosphorus in the Levantine Basin, eastern Mediterranean: Spatial and temporal variability and its role in seawater productivity. *Limnol. Oceanorg.,, 48 (4),* 1557–1568, 2003. doi: 10.4319/lo.2003.48.4.1557

Markaki, Z., Loÿe-Pilot, M. D., Violaki, K., Benyahya, L., Mihalopoulos, N.; Variability of atmospheric deposition of dissolved nitrogen and phosphorus in the Mediterranean and possible link to the anomalous seawater N/P ratio. *Mar. Chem., 120(1–4),* 187–194, 2010. doi: 10.1016/ j.marchem.2008.10.005.

Mihalopoulos, N., Kerminen, V., M., Kanakidou, M., Berresheim, H., Sciare, J.: Formation of particulate sulfur species (sulfate and methanesulfonate) during summer over the Eastern Mediterranean: a modelling approach. *Atmos. Environ. 41,* 6860-6871, 2007.

Miller, J.: The nitrogen content of rain falling at Rothamsted. *J. Agricul. Sci., 1,* 280-303, 1905.

Miyazaki, Y., Kawamura, K., Jung, J., Furutani, H., Uematsu, M.: Latitudinal distributions of organic nitrogen and organic carbon in marine aerosols over the western North Pacific, *Atmos. Chem. Phys., 11,* 3037–3049, 2011. doi:10.5194/acp-11-3037-2011.

Neff, J. C., Holland, E. A., Dentener, F. J., McDowell, W. H., and Russell, K. M.: The origin, composition and rates of organic nitrogen deposition: A missing piece of the nitrogen cycle. *Biogeochemistry, 57*, 99–136, 2002.

Paatero, P., Tapper, U.: Positive matrix factorization: a non-negative factor model with optimal utilization of error estimates of data value. *Environmetrics, 5,* 111–126, 1994.

Peierls, B. L. and Paerl, H. W.: Bioavailability of atmospheric organic nitrogen deposition to coastal phytoplankton. *Limnol. Oceanogr., 42,* 1819–1823, 1997.

Paatero, P.: End User's Guide to Multilinear Engine Applications, 2007. ftp://ftp.clarkson.edu/pub/hopkepk/pmf/.

Paatero, P., Eberly, S., Brown, S. G., Norris, G. A.: Methods for estimating uncertainty in factor analytic solutions. *Atmos. Meas. Tech., 7,* 781–797, 2014. doi:10.5194/amt-7-781-2014.

Powley, H. R., Krom, M. D., Van Cappellen, P.: Understanding the unique biogeochemistry of the Mediterranean Sea: Insights from a coupled phosphorus and nitrogen model. *Global Biogeochem. Cy., 31,* 1010-1031, 2017. doi: 10.1002/2017GB005648.

[revised manuscript text omitted]